# Generative modeling of multi-mapping reads with mHi-C advances analysis of Hi-C studies

Ye Zheng[1], Ferhat Ay[2,3], Sunduz Keles[1,4]*

[1]Department of Statistics, University of Wisconsin-Madison, Madison, United States; [2]La Jolla Institute for Allergy and Immunology, La Jolla, United States; [3]School of Medicine, University of California, San Diego, La Jolla, United States; [4]Department of Biostatistics and Medical Informatics, University of Wisconsin-Madison, Madison, United States

**Abstract** Current Hi-C analysis approaches are unable to account for reads that align to multiple locations, and hence underestimate biological signal from repetitive regions of genomes. We developed and validated *mHi-C*, a *m*ulti-read mapping strategy to probabilistically allocate Hi-C multi-reads. mHi-C exhibited superior performance over utilizing only uni-reads and heuristic approaches aimed at rescuing multi-reads on benchmarks. Specifically, mHi-C increased the sequencing depth by an average of 20% resulting in higher reproducibility of contact matrices and detected interactions across biological replicates. The impact of the multi-reads on the detection of significant interactions is influenced marginally by the relative contribution of multi-reads to the sequencing depth compared to uni-reads, *cis-to-trans* ratio of contacts, and the broad data quality as reflected by the proportion of mappable reads of datasets. Computational experiments highlighted that in Hi-C studies with short read lengths, mHi-C rescued multi-reads can emulate the effect of longer reads. mHi-C also revealed biologically supported *bona fide* promoter-enhancer interactions and topologically associating domains involving repetitive genomic regions, thereby unlocking a previously masked portion of the genome for conformation capture studies.
DOI: https://doi.org/10.7554/eLife.38070.001

*For correspondence:
keles@stat.wisc.edu

**Competing interests:** The authors declare that no competing interests exist.

## Introduction

DNA is highly compressed in the nucleus and organized into a complex three-dimensional structure. This compressed form brings distal functional elements into close spatial proximity of each other (*Dekker et al., 2002*; *de Laat and Duboule, 2013*) and has a far-reaching influence on gene regulation. Changes in DNA folding and chromatin structure remodeling may result in cell malfunction with devastating consequences (*Corradin et al., 2016*; *Won et al., 2016*; *Javierre et al., 2016*; *Rosa-Garrido et al., 2017*; *Spielmann et al., 2018*). Hi-C technique (*Lieberman-Aiden et al., 2009*; *Rao et al., 2014*) emerged as a high throughput technology for interrogating the three-dimensional configuration of the genome and identifying regions that are in close spatial proximity in a genome-wide fashion. Thus, Hi-C data is powerful for discovering key information on the roles of the chromatin structure in the mechanisms of gene regulation.

There are a growing number of published and well-documented Hi-C analysis tools and pipelines (*Heinz et al., 2010*; *Hwang et al., 2015*; *Ay et al., 2014a*; *Servant et al., 2015*; *Mifsud et al., 2015*; *Lun and Smyth, 2015*), and their operating characteristics were recently studied (*Ay and Noble, 2015*; *Forcato et al., 2017*; *Yardımcı et al., 2017*) in detail. However, a key and common step in these approaches is the exclusive use of uniquely mapping reads. Limiting the usable reads to only uniquely mapping reads underestimates signal originating from repetitive regions of the

genome which are shown to be critical for tissue specificity (*Xie et al., 2013*). Such reads from repetitive regions can be aligned to multiple positions (*Figure 1A*) and are referred to as multi-mapping reads or multi-reads for short. The critical drawbacks of discarding multi-reads have been recognized in other classes of genomic studies such as transcriptome sequencing (RNA-seq) (*Li and Dewey, 2011*), chromatin immunoprecipitation followed by high throughput sequencing (ChIP-seq) (*Chung et al., 2011*; *Zeng et al., 2015*), as well as genome-wide mapping of protein-RNA binding sites (CLIP-seq or RIP-seq) (*Zhang and Xing, 2017*). More recently, (*Sun et al., 2018*) and (*Cournac et al., 2016*) argued for a fundamental role of repeat elements in the 3D folding of genomes, highlighting the role of higher order chromatin architecture in repeat expansion disorders. However, the ambiguity of multi-reads alignment renders it a challenge to investigate the repetitive elements co-localization with the true 3D interaction architecture and signals. In this work, we developed mHi-C (*Figure 1—figure supplements 1* and *2*), a hierarchical model that probabilistically allocates Hi-C multi-reads to their most likely genomic origins by utilizing specific characteristics of the paired-end reads of the Hi-C assay. mHi-C is implemented as a full analysis pipeline (https://github.com/keleslab/mHiC) that starts from unaligned read files and produces a set of statistically significant interactions at a given resolution. We evaluated mHi-C both by leveraging replicate structure of public of Hi-C datasets of different species and cell lines across six different studies, and also with computational trimming and data-driven simulation experiments.

## Results

### Multi-reads significantly increase the sequencing depths of Hi-C data

For developing mHi-C and studying its operating characteristics, we utilized six published studies, resulting in eight datasets with multiple replicates, as summarized in *Table 1* and with more details in *Figure 1—source data 1*: Table 1. These datasets represent a variety of study designs from different organisms, that is human and mouse cell lines as examples of large genomes and three different stages of *Plasmodium falciparum* red blood cell cycle as an example of a small and AT-rich genome. Specifically, they span a wide range of sequencing depths (*Figure 1B*), coverages and *cis*-to-*trans* ratios (*Figure 1—figure supplement 3*), and have different proportions of mappable and valid reads (*Figure 1—figure supplement 4*). Before applying mHi-C to these datasets and investigating biological implications, we first established the substantial contribution of multi-reads to the sequencing depth across these datasets with diverse characteristics. At read-end level (*Supplementary file 1* for terminology), after the initial step of aligning to the reference genome (*Figure 1—figure supplements 1* and *2*), multi-reads constitute approximately 10% of all the mappable read ends (*Figure 1—figure supplement 5A*). Moreover, the contribution of multi-reads to the set of aligned reads further increases by an additional 8% when chimeric reads (*Supplementary file 1*) are also taken into account (*Figure 1—source data 1*: Table 2). Most notably, *Figure 1—figure supplement 6A* demonstrates that, in datasets with shorter read lengths, multi-reads constitute a larger percentage of usable reads compared to uniquely mapping chimeric reads that are routinely rescued in Hi-C analysis pipelines (*Servant et al., 2015*; *Lun and Smyth, 2015*; *Durand et al., 2016*). Moreover, multi-reads also make up a significant proportion of the rescued chimeric reads (*Figure 1—figure supplement 6B*). At the read pair level, after joining of both ends, multi-reads increase the sequencing depth by 18% to 23% for shorter read length datasets and 10% to 15% for longer read lengths, thereby providing a substantial increase to the depth before the read pairs are further processed into bin pairs (*Figure 1C*; *Figure 1—source data 1*: Table 3).

### Multi-reads can be rescued at multiple processing stages of mHi-C pipeline

As part of the post-alignment pre-processing steps, Hi-C reads go through a series of validation checking to ensure that the reads that represent biologically meaningful fragments are retained and used for downstream analysis (*Figure 1—figure supplements 1* and *2*, *Supplementary file 1*). mHi-C pipeline tracks multi-reads through these processing steps. Remarkably, in the application of mHi-C to all six studies, a subset of the high-quality multi-reads are reduced to uni-reads either in the validation step when only one candidate contact passes the validation screening, or because all the alignments of a multi-read reside within the same bin (*Figure 1C*; *Supplementary file 1* and see

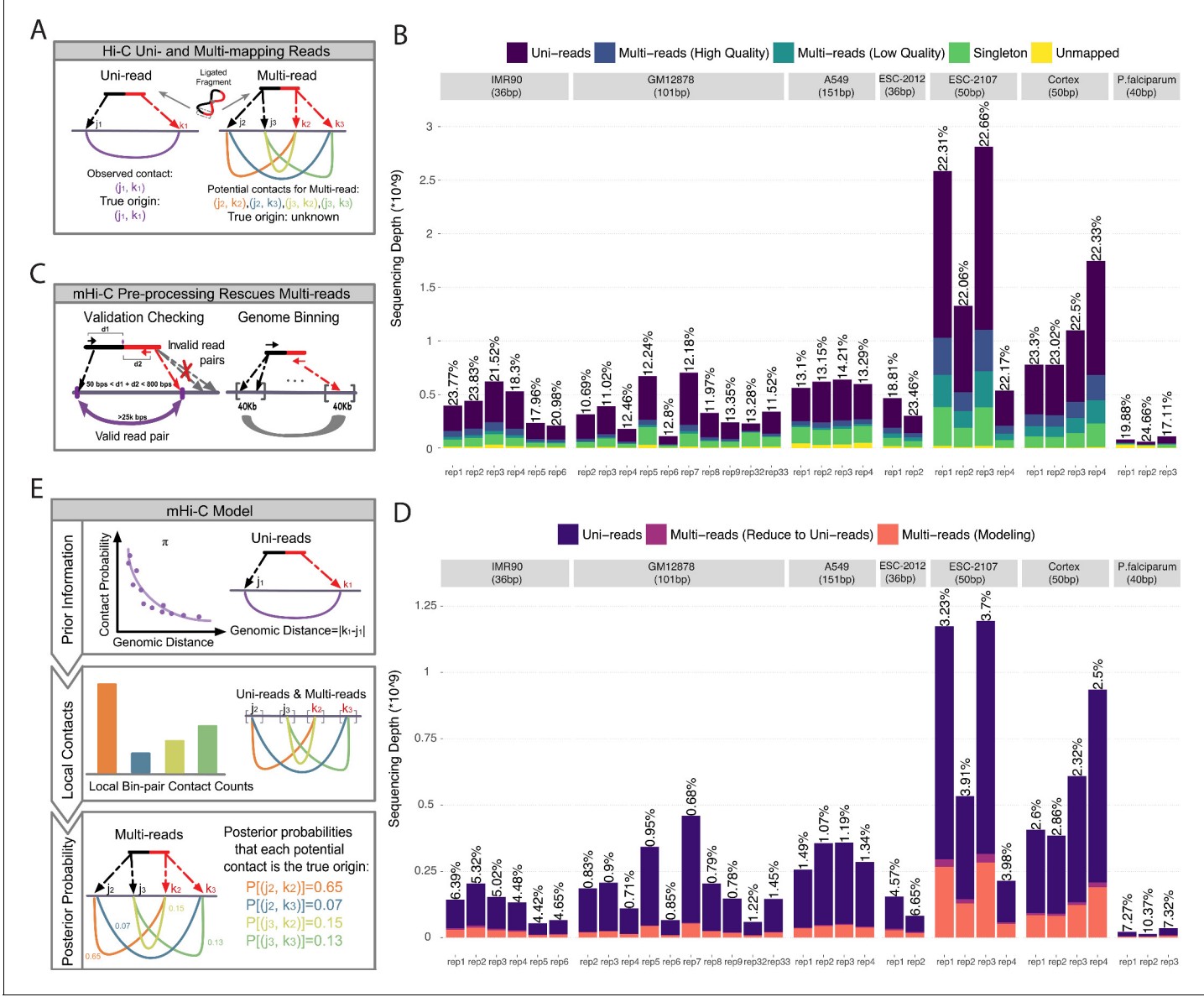

**Figure 1.** Overview of multi-reads and mHi-C pipeline. (**A**) Standard Hi-C pipelines utilize uni-reads while discarding multi-mapping reads which give rise to multiple potential contacts. (**B**) The total number of reads in different categories as a result of alignment to reference genome across the study datasets. Percentages of high-quality multi-reads compared to uni-reads are depicted on top of each bar. (**C**) Multi-mapping reads can be reduced to uni-reads within validation checking and genome binning pre-processing steps. (**D**) Aligned reads after validation checking and binning. Percentage improvements in sequencing depths due to multi-reads becoming uni-reads are depicted on top of each bar. (**E**) mHi-C modeling starts from the prior built by only uni-reads to quantify the relationship between random contact probabilities and the genomic distance between the contacts. This prior is updated by leveraging local bin pair contacts including both uni- and multi-reads and results in posterior probabilities that quantify the evidence for each potential contact to be the true genomic origin.

DOI: https://doi.org/10.7554/eLife.38070.002

The following source data and figure supplements are available for figure 1:

**Source data 1.** Detailed summary of study datasets.
DOI: https://doi.org/10.7554/eLife.38070.009
**Figure supplement 1.** mHi-C pipeline (Alignment - Read end pairing - Valid fragment filtering).
DOI: https://doi.org/10.7554/eLife.38070.003
**Figure supplement 2.** mHi-C pipeline (Duplicate removal - Genome binning - mHi-C).
DOI: https://doi.org/10.7554/eLife.38070.004
**Figure supplement 3.** Coverage and *cis*-to-*trans* ratios across individual replicates of the study datasets as indicators of data quality.
*Figure 1 continued on next page*

*Figure 1 continued*

DOI: https://doi.org/10.7554/eLife.38070.005

**Figure supplement 4.** Percentages of (A) mappable and (B) valid reads over the set of all reads for individual replicates of the study datasets as an indicator of data quality.

DOI: https://doi.org/10.7554/eLife.38070.006

**Figure supplement 5.** Categorization of reads after alignment across study datasets.

DOI: https://doi.org/10.7554/eLife.38070.007

**Figure supplement 6.** Comparison of the prevalence of multi-reads and chimeric reads, both of which require additional processing.

DOI: https://doi.org/10.7554/eLife.38070.008

Materials and methods). Collectively, mHi-C can rescue as high as 6.7% more valid read pairs (*Figure 1D*) that originate from multi-reads and are mapped unambiguously without carrying out any multi-reads specific procedure for large genomes and 10.4% for *P. falciparum*. Such improvement corresponds to millions of reads for deeper sequenced datasets (*Figure 1—source data 1*: Table 4). For the remaining multi-reads (*Figure 1D*, colored in pink), which, on average, make up 18% of all the valid reads (*Figure 1—figure supplement 5B*), mHi-C implements a novel multi-mapping model and probabilistically allocates them.

mHi-C generative model (*Figure 1E* and see Materials and methods) is constructed at the bin-pair level to accommodate the typical signal sparsity of genomic interactions. The bins are either fixed-size non-overlapping genome intervals or a fixed number of restriction fragments derived from the Hi-C protocol. The resolutions at which seven cell lines are processed are summarized in *Table 1*. In the mHi-C generative model, we denote the observed alignment indicator vector for a given paired-end read $i$ by vector $Y_i$ and use unobserved hidden variable vector $Z_i$ to indicate its true genomic origin. Contacts captured by Hi-C assay can arise as random contacts of nearby genomic positions or true biological interactions. mHi-C generative model acknowledges this feature by utilizing data-driven priors, $\pi_{(j,k)}$ for bin pairs $j$ and $k$, as a function of contact distance between the two bins. mHi-C updates these prior probabilities for each candidate bin pair that a multi-read can be allocated to by leveraging local contact counts. As a result, for each multi-read $i$, it estimates posterior probabilities of genomic origin variable $Z_i$. Specifically, $Pr(Z_{i,(j,k)} = 1 \,|Y_i,\, \pi)$ denotes the posterior probability, that is allocation probability, that the two read ends of multi-read $i$ originate from bin pairs $j$ and $k$. These posterior probabilities, which can also be viewed as fractional contacts of multi-read $i$, are then utilized to assign each multi-read to the most likely genomic origin. Our results in this paper only utilized reads with allocation probability greater than 0.5. This ensured the output of mHi-C to be compatible with the standard input of the downstream normalization and statistical significance estimation methods (*Imakaev et al., 2012*; *Knight and Ruiz, 2013*; *Ay et al., 2014a*).

**Table 1.** Hi-C Data Summary.

| Cell line | Replicate | Read length (bp) | Restriction enzyme | HiC protocol | Source | Resolution (kb) |
|---|---|---|---|---|---|---|
| IMR90 | rep1-6 | 36 | HindIII | dilution | (*Jin et al., 2013*) | 40 |
| GM12878 | rep2-9 | 101 | MboI | in situ | (*Rao et al., 2014*) | 5, 10*, 40* |
| GM12878 | rep32, rep33 | 101 | DpnII | in situ | (*Rao et al., 2014*) | 5 |
| A549 | rep1-4 | 151 | MboI | in situ | (*Dixon et al., 2018*) | 10, 40 |
| ESC(2012) | rep1, rep2 | 36 | HindIII | dilution | (*Dixon et al., 2012*) | 40 |
| ESC(2017) | rep1-4 | 50 | DpnII | in situ | (*Bonev et al., 2017*) | 10, 40 |
| Cortex | rep1-4 | 50 | DpnII | in situ | (*Bonev et al., 2017*) | 10, 40 |
| *P. falciparum* | three stages | 40 | MboI | dilution | (*Ay et al., 2014b*) | 10, 40 |

*Replicates 2, 3, 4, and 6 of the GM12878 cell line datasets were process at 10 kb and 40 kb resolutions.

DOI: https://doi.org/10.7554/eLife.38070.010

## Probabilistic assignment of multi-reads results in more complete contact matrices and significantly improves reproducibility across replicates

Before quantifying mHi-C model performance, we provide a direct visual comparison of the contact matrices between Uni-setting and Uni&Multi-setting using raw and normalized contact counts. *Figure 2A* and *Figure 2—figure supplements 1–4* clearly illustrate how multi-mapping reads fill in the low mappable regions and lead to more complete matrices, corroborating that repetitive genomic regions are under-represented without multi-reads. Quantitatively, for the combined replicates of GM12878, 99.61% of the 5 kb bins with interaction potential are covered by at least 100 raw contacts under the Uni&Multi-setting, compared to 98.72% under Uni-setting, thereby allowing us to study 25.55 Mb more of the genome. For normalized contact matrices, the coverage increases from 99.42% in Uni-setting to 99.97% in Uni&Multi-setting (*Figure 2—figure supplement 5*). In addition to increasing the sequencing depth in extremely low contact bins for both raw and normalized contact counts, higher bin-level coverage after leveraging multi-mapping reads appears as a global pattern across the genome for raw contact matrices (*Figure 2—figure supplements 6* and *7*). *Figure 2—figure supplement 8* provides the histogram of bin-level differences of normalized contact counts between the two settings and indicates a positive average difference. While some bins appear to have their contact counts decreased in the Uni&Multi-setting compared to Uni-setting after normalization (purple bar in *Figure 2—figure supplement 8A*), comparison of the raw contact counts in *Figure 2—figure supplement 8B* shows that these bins do indeed have lower raw contact counts in the Uni-setting compared to Uni&Multi-setting and that the reduction observed is an artifact of normalization. This also highlights that multi-reads alleviate the inflation of low raw contact count regions due to normalization. These major improvements in coverage provide direct evidence that mHi-C is rescuing multi-reads that originate from biologically valid fragments.

We assessed the impact of multi-reads rescued by mHi-C on the reproducibility from the point of both raw contact counts and significant interactions detected. We used the stratum-adjusted correlation coefficient proposed in HiCRep (*Yang et al., 2017*) for evaluating the reproducibility of Hi-C contact matrices. *Figure 2B* and *Figure 2—figure supplements 9* and *10* illustrate that integrating multi-reads leads to increased reproducibility and reduced variability of stratum-adjusted correlation coefficients among biological replicates across all the study datasets. Furthermore, we observe that, for some chromosomes, for example, chr17 of IMR90 and chr16 of GM12878, the improvement in reproducibility stands out, without a systematic behavior across datasets. A close examination of improvement in reproducibility as a function of the ratio of rescued multi-reads to uni-reads across chromosomes highlights the larger proportion of multi-reads rescued for these chromosomes (*Figure 2—figure supplement 11*). To further assess that the improvement in reproducibility did not manifest due to an unaccounted systematic bias in the assignment of multi-reads, we evaluated reproducibility similarly between replicates of GM12878 and replicates of IMR90. *Figure 2—figure supplement 12* shows that Uni-setting and Uni&Multi-setting lead to similar levels of reproducibility between replicates of these unrelated samples with all Wilcoxon rank-sum test p-values of the pairwise comparisons between Uni- and Uni&Multi-settings > 0.21; therefore, ruling out the possibility of a systematic bias as the source of improvement in reproducibility due to multi-reads.

In addition to the direct comparison of the raw contact matrices and their reproducibility, we identified the set of significant interactions by Fit-Hi-C (*Ay et al., 2014a*) and assessed the reproducibility of the identified interactions. *Figure 2C* shows that mHi-C significantly improves reproducibility of detected interactions across all the pairwise comparisons of replicates within each study dataset. *Figure 2—figure supplement 13A* presents more details on the degree of overlap among the significant interactions identified at 5% and 10% false discovery rate (FDR) across replicates for the IMR90 datasets. These comparisons highlight that significant interactions specific to Uni&Multi-setting have consistently higher reproducibility than those specific to Uni-setting across all pairwise comparisons. Since random contacts tend to arise due to short genomic distances between loci, we stratified the significant interactions based on distance and reassessed the reproducibility as a function of the genomic distance between the contacts (*Figure 2—figure supplement 13B*). Notably, significant interactions identified only by the Uni-setting and those common to both settings have a stronger gradual descending trend as a function of the genomic distance, indicating decaying

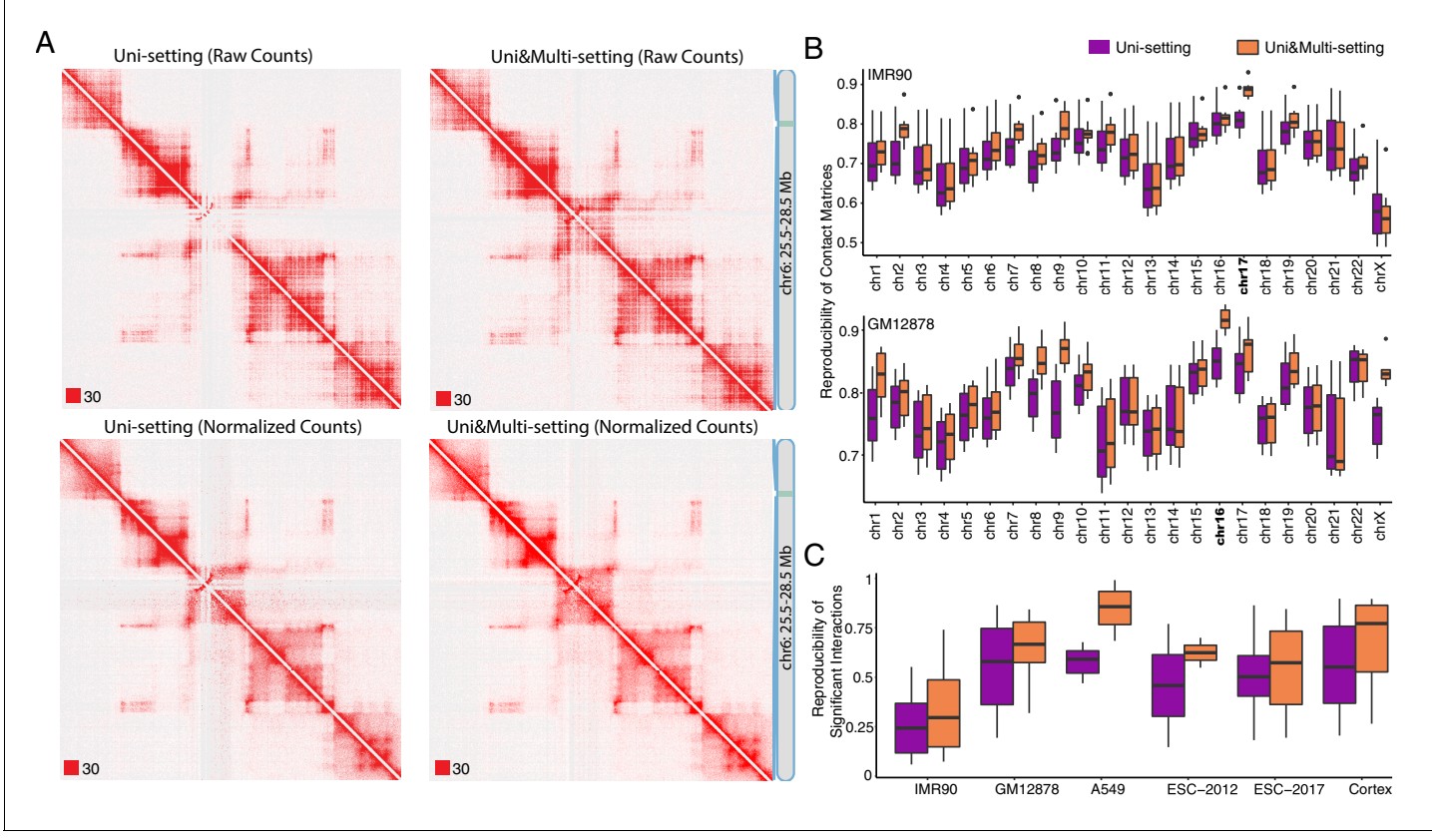

**Figure 2.** Global impact of multi-reads in Hi-C analysis. (**A**) Contact matrices of GM12878 with combined reads from replicates 2–9 are compared under Uni-setting and Uni&Multi-setting using raw and normalized contact counts for chr6:25.5 Mb - 28.5 Mb. White gaps of Uni-reads contact matrix, due to lack of reads from repetitive regions, are filled in by multi-reads, hence resulting in a more complete contact matrix. Such gaps remain in the Uni-setting even after normalization. Red squares at the left bottom of the matrices indicate the color scale. (**B**) Reproducibility of Hi-C contact matrices by HiCRep across all pairwise comparisons between replicates under the Uni- and Uni&Multi-settings (IMR90 and GM12878 are displayed). (**C**) Reproducibility of the significant interactions across replicates of the study datasets. Reproducibility is assessed by overlapping interactions detected at FDR of 5% for pairs of replicates within each study dataset.

DOI: https://doi.org/10.7554/eLife.38070.011

The following figure supplements are available for figure 2:

**Figure supplement 1.** Raw and normalized contact matrices of GM12878 under Uni-setting and Uni&Multi-setting on chromosome 1.
DOI: https://doi.org/10.7554/eLife.38070.012

**Figure supplement 2.** Raw and normalized contact matrices of GM12878 under Uni-setting and Uni&Multi-setting on chromosome 2.
DOI: https://doi.org/10.7554/eLife.38070.013

**Figure supplement 3.** Raw and normalized contact matrices of GM12878 under Uni-setting and Uni&Multi-setting on chromosome 3.
DOI: https://doi.org/10.7554/eLife.38070.014

**Figure supplement 4.** Raw and normalized contact matrices of GM12878 under Uni-setting and Uni&Multi-setting on chromosome 5.
DOI: https://doi.org/10.7554/eLife.38070.015

**Figure supplement 5.** Proportion of bins that are covered by at least 100 (row 1) or 1000 (row 2) contacts for raw contact matrices (column 1) and normalized contact matrices (column2) under Uni- and Uni&Multi-settings for GM12878 with combined reads from replicates 2–9 at 5 kb resolution.
DOI: https://doi.org/10.7554/eLife.38070.016

**Figure supplement 6.** Bin coverage improvement of raw contact matrices under Uni&Multi-setting compared to Uni-setting for GM12878 with combined reads from replicates 2–9 at 5 kb.
DOI: https://doi.org/10.7554/eLife.38070.017

**Figure supplement 7.** Bin coverage improvement of raw contact matrices under Uni&Multi-setting compared to Uni-setting for IMR90 at the individual replicate level for two different allocation probability thresholds.
DOI: https://doi.org/10.7554/eLife.38070.018

**Figure supplement 8.** Bin coverage comparison of normalized contact matrices under Uni&Multi- and Uni-settings for GM12878 with combined reads from replicates 2–9 at 5 kb.
DOI: https://doi.org/10.7554/eLife.38070.019

*Figure 2 continued on next page*

*Figure 2 continued*

**Figure supplement 9.** Reproducibility at the contact matrix level under the Uni- and Uni&Multi-settings in A549, ESC-2017 and Cortex cell lines.
DOI: https://doi.org/10.7554/eLife.38070.020
**Figure supplement 10.** Reproducibility at the contact matrix level at resolutions 40 kb (low) and 10 kb (high) across study datasets.
DOI: https://doi.org/10.7554/eLife.38070.021
**Figure supplement 11.** Percent improvement in reproducibility due to the Uni&Multi-setting versus the proportion of the number of valid multi-reads compared to the number of the uni-reads in the datasets.
DOI: https://doi.org/10.7554/eLife.38070.022
**Figure supplement 12.** Reproducibility at the contact matrix level under the Uni- and Uni&Multi-settings between GM12878 and IMR90 at 40 kb resolution.
DOI: https://doi.org/10.7554/eLife.38070.023
**Figure supplement 13.** Reproducibility of significant interactions for IMR90.
DOI: https://doi.org/10.7554/eLife.38070.024

reproducibility for long-range interactions. In contrast, Uni&Multi-setting maintains a relatively higher and stable reproducibility for longer genomic distances.

## 2.4 Multi-reads detect novel significant interactions

At 5% false discovery rate, mHi-C detects 20% to 50% more novel significant interactions for relatively highly sequenced study datasets (*Figure 3A* and *Figure 3—source data 1*; *Figure 3—figure supplement 1* for other FDR thresholds and resolutions). The gains are markedly larger for datasets with smaller sequencing depths (e.g., ESC-2012) or extremely high coverage (e.g., *P. falciparum*). Overall gains in the number of novel contacts persist as the cutoff for mHi-C posterior probabilities of multi-read assignments varies (*Figure 3—figure supplement 2*). At fixed FDR, significant interactions identified by the Uni&Multi-setting also include the majority of significant interactions inferred from the Uni-setting, indicating that incorporating multi-reads is extending the significant interaction list (low level of purple lines in *Figure 3—figure supplement 2*).

We leveraged the diverse characteristics of the study datasets and investigated the factors that impacted the gain in the detected significant interactions due to multi-reads. The top row of *Figure 3—figure supplement 3* summarizes the marginal correlations of the percentage change in the number of identified significant interactions (at 40 kb resolution and FDR of 0.05) with the data characteristics commonly used to indicate the quality of Hi-C datasets (excluding the high coverage *P. falciparum* dataset). These marginal associations highlight the significant impact of the relative contribution of multi-reads to the sequencing depth compared to uni-reads and *cis*-to-*trans* ratio of contacts (*Figure 3—figure supplement 4*). *Figure 3—figure supplement 5* increase in the number of novel significant interactions for the GM12878 datasets in more detail across a set of FDR thresholds and at different resolutions, and includes two types of restriction enzymes. Specifically, *Figure 3—figure supplement 5C* illustrates a clear negative association between the sequencing depth and the percent improvement in the number of identified significant interactions at 5 kb resolution due to the larger impact of multi-reads on the smaller depth replicates. As an exception, we note that *P. falciparum* datasets tend to exhibit significantly higher gains in the number of identified contacts especially under stringent FDR thresholds (*Figure 3A*), possibly due to the ultra-high coverage of these datasets (*Figure 3—figure supplement 6*). In addition to these marginal associations, *Figure 3B* and *Figure 3—figure supplement 7* display the percentage increase in the number of identified significant interactions as a function of the percentage increase in the real depth due to multi-reads and the *cis*-to-*trans* ratio across all the study datasets. A consistent pattern highlights that short read datasets with large proportion of mHi-C rescued multi-reads compared to uni-reads enjoy a larger increase in the number of identified significant interactions regardless of the FDR threshold, while for datasets with similar relative contribution of multi-reads, for example within lower depth IMR90, *cis*-to-*trans* ratios positively correlate with the increase in the number of identified significant interactions.

We next asked whether novel significant interactions due to rescued multi-reads could have been identified under the Uni-setting by employing a more liberal FDR threshold. Leveraging multi-reads with posterior probability larger than 0.5 and controlling the FDR at 1%, Fit-Hi-C identified 32.49% more significant interactions compared to Uni-setting (comparing dark green to dark purple bar in

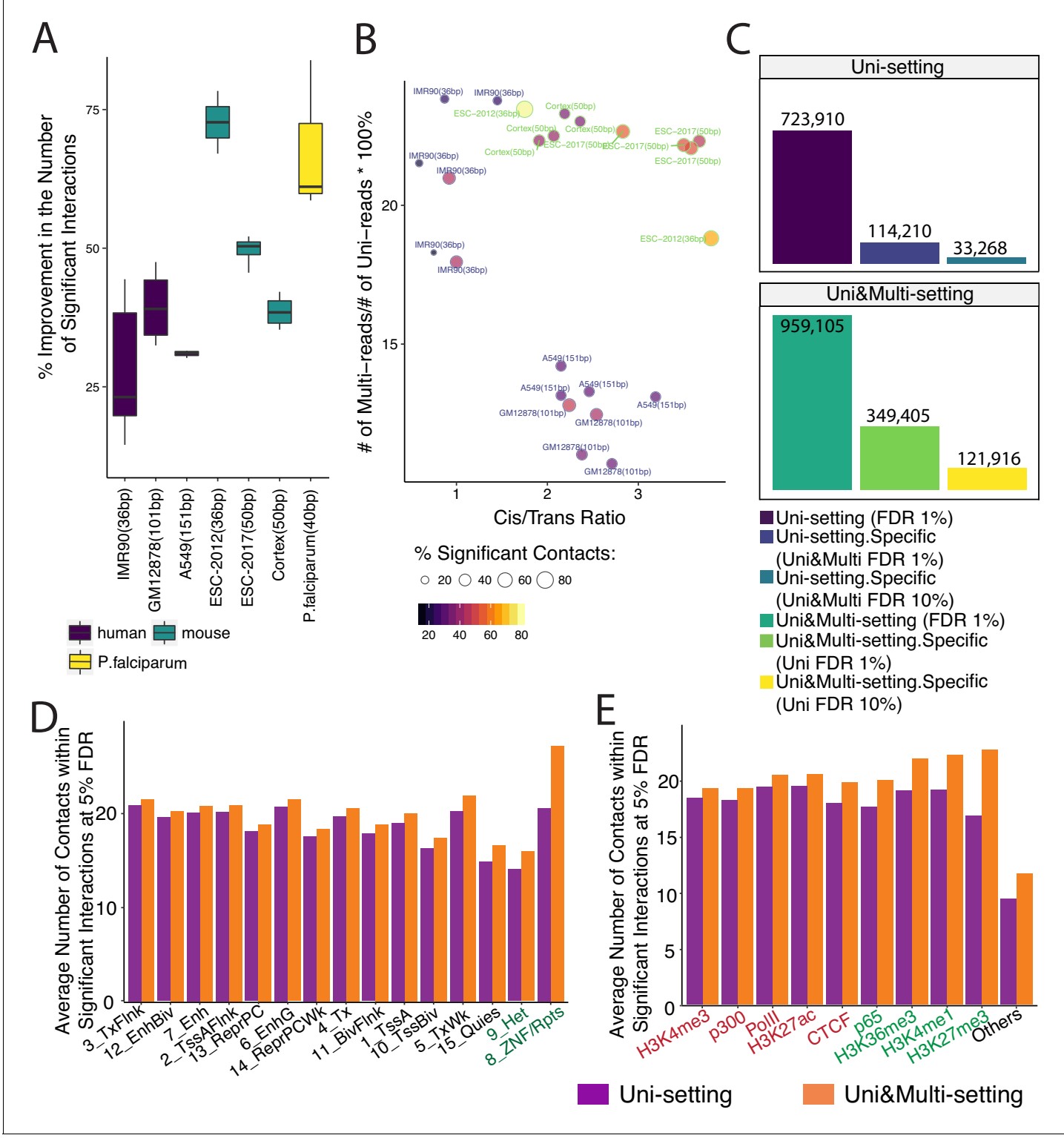

**Figure 3.** Gain in the numbers of novel significant interactions by mHi-C and their characterization by chromatin marks. (**A**) Percentage increase in detected significant interactions (FDR 5%) by comparing contacts identified in Uni&Multi-setting with those of Uni-setting across study datasets at 40 kb resolution. (**B**) Percentage change in the numbers of significant interactions (FDR 5%) as a function of the percentage of mHi-C rescued multi-reads in comparison to uni-read and *cis-to-trans* ratios of individual datasets at 40 kb resolution. (**C**) Recovery of significant interactions identified at 1% FDR by analysis at 10% FDR, aggregated over the replicates of GM12878 at 40 kb resolution. Detailed descriptions of the groups are provided in *Figure 3— figure supplement 6*. (**D**) Average number of contacts falling within the significant interactions (5% FDR) that overlapped with each chromHMM

*Figure 3 continued on next page*

*Figure 3 continued*

annotation category across six replicates of IMR90 identified by Uni- and Uni&Multi-settings. (E) Average number of contacts (5% FDR) that overlapped with significant interactions and different types of ChIP-seq peaks associated with different genomic functions (IMR90 six replicates). Red/Green labels denote smaller/larger differences between the two settings compared to the differences observed in the "Others' category that depict non-peak regions.

DOI: https://doi.org/10.7554/eLife.38070.025

The following source data and figure supplements are available for figure 3:

**Source data 1.** Percentage of improvement in the number of significant interactions across six studies at resolution 40 kb.
DOI: https://doi.org/10.7554/eLife.38070.043

**Figure supplement 1.** Percentage change in the numbers of significant interactions under the Uni&Multi-setting compared to Uni-setting at 0.1%, 1%, 5% and 10% FDR thresholds and resolutions (A) 40 kb and (B) 10 kb.
DOI: https://doi.org/10.7554/eLife.38070.026

**Figure supplement 2.** Comparison of significant interactions as a function of posterior probabilities of multi-read assignment (IMR90 40 kb).
DOI: https://doi.org/10.7554/eLife.38070.027

**Figure supplement 3.** Heatmap for marginal correlations of percentage increase in the number of identified significant interactions (FDR 5%) with indicators of data quality across study datasets excluding *P. falciparum* at 40 kb.
DOI: https://doi.org/10.7554/eLife.38070.028

**Figure supplement 4.** Percentage change in the numbers of significant interactions with respect to *cis-to-trans* ratio excluding *P. falciparum* at 40 kb.
DOI: https://doi.org/10.7554/eLife.38070.029

**Figure supplement 5.** Percentage change in the numbers of significant interactions of GM12878 datasets at different resolutions.
DOI: https://doi.org/10.7554/eLife.38070.040

**Figure supplement 6.** Percentage change in the numbers of significant interactions with respect to coverage at 40 kb.
DOI: https://doi.org/10.7554/eLife.38070.041

**Figure supplement 7.** Percentage change in the numbers of significant interactions as a function of the percentage of mHi-C rescued multi-reads in comparison to uni-reads and *cis-to-trans* ratios at 40 kb.
DOI: https://doi.org/10.7554/eLife.38070.030

**Figure supplement 8.** Recovery of significant interactions identified at FDR 1% by analysis at FDR 10% for each of six replicates of IMR90 at 40 kb.
DOI: https://doi.org/10.7554/eLife.38070.031

**Figure supplement 9.** Recovery of significant interactions identified at FDR 1% by analysis at FDR 10% for each of four replicates of GM12878 at 40 kb resolution.
DOI: https://doi.org/10.7554/eLife.38070.032

**Figure supplement 10.** Recovery of significant interactions identified at FDR 1% by analysis at FDR 10% for each of four replicates of GM12878 at 10 kb resolution.
DOI: https://doi.org/10.7554/eLife.38070.033

**Figure supplement 11.** Recovery of significant interactions identified at FDR 1% by analysis at FDR 10% for each of ten replicates of GM12878 at 5 kb resolution.
DOI: https://doi.org/10.7554/eLife.38070.034

**Figure supplement 12.** Recovery of significant interactions identified at FDR 1% by analysis at FDR 10% for GM12878 summed across replicates at 5 kb, 10 kb, and 40 kb resolutions.
DOI: https://doi.org/10.7554/eLife.38070.042

**Figure supplement 13.** ROC and PR curves for replicates 5 and 6 of IMR90.
DOI: https://doi.org/10.7554/eLife.38070.035

**Figure supplement 14.** Quantification of significant interactions for chromHMM states and ChIP-seq peak regions (IMR90).
DOI: https://doi.org/10.7554/eLife.38070.036

**Figure supplement 15.** Marginalized Hi-C signal (contact counts aggregated across the genomic coordinates for six replicates of IMR90), ChIP-seq coverage and peaks and gene expression for chr1:16,000,000–18,000,000.
DOI: https://doi.org/10.7554/eLife.38070.037

**Figure supplement 16.** Marginalized Hi-C signal (contact counts aggregated across the genomic coordinates for six replicates of IMR90), ChIP-seq coverage and peaks and gene expression for chr2:113460,000–116,000,000.
DOI: https://doi.org/10.7554/eLife.38070.038

**Figure supplement 17.** Marginalized Hi-C signal (contact counts aggregated across the genomic coordinates for six replicates of IMR90), ChIP-seq coverage and peaks and gene expression for chr9:66,250,000–66,950,000.
DOI: https://doi.org/10.7554/eLife.38070.039

*Figure 3C*) and 36.43% of all significant interactions are unique to Uni&Multi-setting (light green bar over dark green bar in *Figure 3C*) collectively for all the four replicates of GM12878 at 40 kb resolution. We observed that 34.89% of these novel interactions (yellow bar over the light green bar in *Figure 3C*) at 1% FDR (i.e., 12.71% compared to the all the significant interactions under Uni&Multi-setting) cannot be recovered even by a more liberal significant interaction list under Uni-setting at 10% FDR. Conversely, Uni&Multi-setting is unable to recover only 4.60% of the Uni-setting contacts once the FDR is controlled at 10% for the Uni&Multi-setting (light blue over dark purple bar in *Figure 3C*), highlighting again that Uni&Multi-setting predominantly adds on novel significant interactions while retaining interactions that are identifiable under the Uni-setting. A similar analysis for individual replicates of IMR90 are provided in *Figure 3—figure supplement 8* as well as those of GM12878 at the individual replicate level or collective analysis at 5 kb, 10 kb, and 40 kb resolutions in *Figure 3—figure supplements 9–12*. We further confirmed this consistent power gain by a Receiver Operating Characteristic (ROC) and a Precision-Recall (PR) analysis (*Figure 3—figure supplement 13*). The PR curve illustrates that at the same false discovery rate (1-precision), mHi-C achieves consistently higher power (recall) than the Uni-setting in addition to better AUROC performance.

## Chromatin features of novel significant interactions

To further establish the biological implications of mHi-C rescued multi-reads, we investigated genomic features of novel contacts. Annotation of the significant interactions with ChromHMM segmentations from the Roadmap Epigenomics project (*Kundaje et al., 2015*) highlights marked enrichment of significant interactions in annotations involving repetitive DNA (*Figure 3D*, *Figure 3—figure supplement 14A*). Most notably, ZNF genes and repeats and Heterochromatin states exhibit the largest discrepancy of the average significant interaction counts between the Uni- and Uni&Multi-settings. To complement the evaluation with ChromHMM annotations, we evaluated the Uni-setting and Uni&Multi-setting significant interaction enrichment of genomic regions harboring histone marks and other biochemical signals (*ENCODE Project Consortium, 2012*; *Jin et al., 2013*) (See Materials and methods) by comparing their average contact counts to those without such signal (*Figure 3E* and data on Dryad, https://doi.org/10.5061/dryad.v7k3140). Notably, while we observe that multi-reads boost the average number of contacts with biochemically active regions of the genome, they contribute more to regions that harbor H3K27me3 peaks (*Figure 3E*, *Figure 3—figure supplement 14B*). Such regions are associated with downregulation of nearby genes through forming heterochromatin structure (*Ferrari et al., 2014*). *Figure 3—figure supplement 15–17* further provide specific examples of how increased marginal contact counts due to multi-reads are supported by signals of histone modifications, CTCF binding sites, and gene expression. Many genes of biological significance reside in these regions. For example, NBPF1 (*Figure 3—figure supplement 15*) is implicated in many neurogenetic diseases and its family consists of dozens of recently duplicated genes primarily located in segmental duplications (*Safran et al., 2010*). In addition, RABL2A within the highlighted region of *Figure 3—figure supplement 16* is a member of RAS oncogene family.

## Multi-reads discover novel promoter-enhancer interactions

We found that a significant impact of multi-reads is on the detection of promoter-enhancer interactions. Overall, mHi-C identifies 14.89% more significant promoter-enhancer interactions at 5% FDR collectively for six replicates for IMR90 (*Figure 4—source data 1*: Table 1 and *Figure 4—source data 2*). Of these interactions, 13,313 are reproducible among all six replicates under Uni&Multi-setting (*Figure 4—source data 1*: Table 2) and 62,971 are reproducible for at least two replicates (*Figure 4—source data 1*: Table 3) leading to 15.84% more novel promoter-enhancer interactions specific to Uni&Multi-setting. *Figure 4A* provides WashU epigenome browser (*Zhou et al., 2011*) display of such novel reproducible promoter-enhancer interactions on chromosome 1. *Figure 4—figure supplements 1–2* provides more such reproducible examples and *Figure 4—figure supplement 3* depicts the reproducibility of these interactions in more details across the six replicates.

We next validated the novel promoter-enhancer interactions by investigating the expression levels of the genes contributing promoters to these interactions. *Figure 4B* supports that genes with significant interactions in their promoters generally exhibit higher expression levels (comparing bars 1–5 to bars 8–9 in *Figure 4B*). Furthermore, if these interactions involve an enhancer, the average

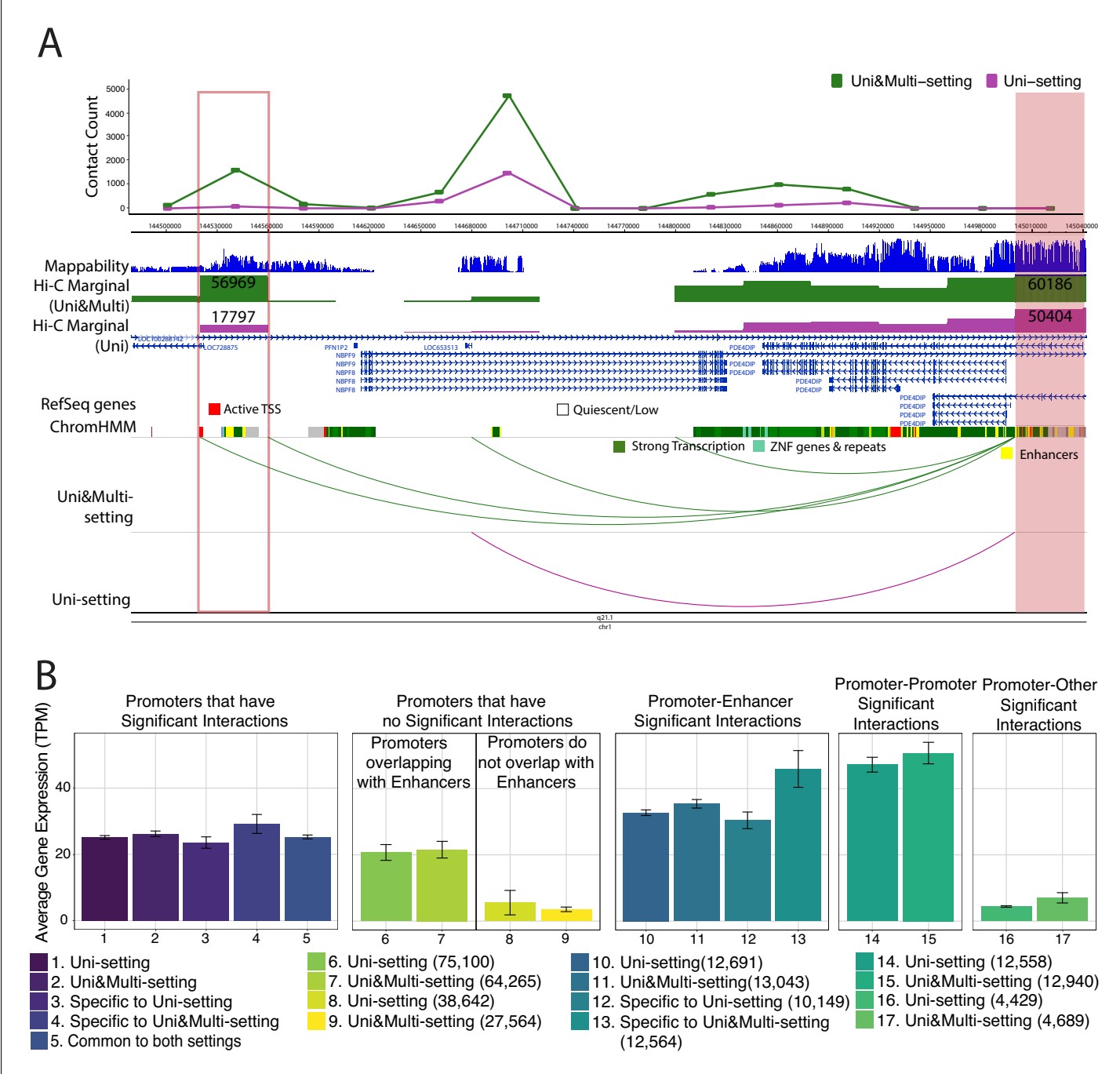

**Figure 4.** Novel promoter-enhancer interactions are reproducible and associated with actively expressed genes. (**A**) mHi-C identifies novel significant promoter-enhancer interactions (green arcs) that are reproducible among at least two replicates in addition to those reproducible under the Uni-setting (purple arcs). Shaded and the boxed regions correspond to the anchor and target bins, respectively. The top track displays the contact counts associated with the anchor bin under Uni- and Uni&Multi-settings. Related chromHMM annotation color labels are added around the track. The complete color labels are consistent with ChromHMM 15-state model at https://egg2.wustl.edu/roadmap/web_portal/chr_state_learning.html. (**B**) Average gene expression with standard errors for five different scenarios of interactions that group promoters into six different categories. In the first panel, significant interactions involving promoters are classified into five settings, and the average gene expressions across genes with the corresponding promoters are depicted. The second panel involves two alignment settings and genes without any promoter interactions at 5% FDR. This panel is further separated into two categories: promoters that overlap with enhancer annotated regions and those that do not. The latter one serves as the baseline for average expression. Genes contributing to the third and fourth panel have promoter-enhancer, promoter-promoter interactions at 5% FDR. The fifth panel considers genes promoters of which have significant interactions with non-enhancer and non-promoter regions. Numbers in the parenthesis correspond to the number of transcripts in each category.

*Figure 4 continued on next page*

*Figure 4 continued*

DOI: https://doi.org/10.7554/eLife.38070.045

The following source data and figure supplements are available for figure 4:

**Source data 1.** The number of significant promoter-enhancer Hi-C interactions at FDR 5% under Uni-setting and Uni&Multi-setting, respectively, for six replicates of IMR90.
DOI: https://doi.org/10.7554/eLife.38070.049

**Source data 2.** Significant promoter-enhancer interactions at FDR 5% under Uni-setting and Uni&Multi-setting for six replicates of IMR90 with the number of contacts.
DOI: https://doi.org/10.7554/eLife.38070.050

**Figure supplement 1.** Examples of significant promoter-enhancer interactions reproducible among six replicates under Uni- and Uni&Multi-settings (IMR90) on chromosome 7.
DOI: https://doi.org/10.7554/eLife.38070.046

**Figure supplement 2.** Examples of significant promoter-enhancer interactions reproducible among 6 replicates under Uni- and Uni&Multi-settings (IMR90) on chromosome 17.
DOI: https://doi.org/10.7554/eLife.38070.051

**Figure supplement 3.** Significant promoter-emhancer interactions under Uni- and Uni&Multi-settings across 6 IMR90 replicates (Chromosome 17).
DOI: https://doi.org/10.7554/eLife.38070.047

**Figure supplement 4.** Expression distribution of genes promoters of which have significant promoter interactions (IMR90).
DOI: https://doi.org/10.7554/eLife.38070.048

gene expression can be 38.17% higher than that of the overall promoters with significant interactions (comparing bars 10–11 to bars 1–2 in *Figure 4B*). Most remarkably, newly detected significant promoter-enhancer interactions (bar 13 in *Figure 4B*) exhibit a stably higher gene expression level, highlighting that, without multi-reads, biologically supported promoter-enhancer interactions are underestimated. In addition, an overall evaluation of significant interactions (5% FDR) that considers interactions from promoters with low expression (TPM $\leq$ 1) as false positives illustrate that mHi-C specific significant promoter interactions have false positive rates comparable to or smaller than those of significant promoter interactions common to Uni- and Uni&Multi-settings (*Figure 4—figure supplement 4*). In contrast, Uni-setting specific interactions have elevated false positive rates.

## Multi-reads refine the boundaries of topologically associating domains

We next investigated the impact of mHi-C rescued multi-reads on the topologically associating domains (TADs) (*Pombo and Dillon, 2015*), where we used a broad definition of TADs to include contact and loop domains. We used the DomainCaller (*Dixon et al., 2012*; *Dixon et al., 2015*) to infer TADs of IMR90 datasets at 40 kb resolution and Arrowhead (*Rao et al., 2014*) for GM12878 datasets at 5 kb resolution under Uni&Multi-settings (*Figure 5—source data 1* and *2*). The detected TADs are compared to those under the Uni-setting. While this comparison did not reveal stark differences in the numbers of TADs identified under the two settings (*Figure 5—figure supplement 1*), we found that Uni&Multi-setting identifies 2.01% more reproducible TADs with 2.36% lower non-reproducible TADs across replicates (*Figure 5A*). Several studies have revealed the role of CTCF in establishing the boundaries of genome architecture (*Ong and Corces, 2014*; *Tang et al., 2015*; *Hsu et al., 2017*). While this is an imperfect indicator of TAD boundaries, we observed that a slightly higher proportion of the detected TADs have CTCF peaks with convergent CTCF motif pairs at the boundaries once multi-reads are utilized (*Figure 5—figure supplement 2A–C*). *Figure 5B* provides an explicit example of how the gap in the contact matrix due to depletion of multi-reads biases the inferred TAD structure. In addition to discovery of novel TADs (*Figure 5—figure supplement 3*) by filling in the gaps in the contact matrix and boosting the domain signals, mHi-C also refines TAD boundaries (*Figure 5—figure supplements 4* and *5*), and eliminates potential false positive TADs that are split by the contact depleted gaps in Uni-setting (*Figure 5—figure supplements 6–8*). The novel, adjusted, and eliminated TADs are largely supported by CTCF signal identified using both uni- and multi-reads ChIP-seq datasets (*Zeng et al., 2015*) as well as convergent CTCF motifs (*Figure 5—figure supplement 2D*), providing support for mHi-C driven modifications to these TADs and revealing a slightly lower false discovery rate for mHi-C compared to Uni-setting (*Figure 5C*, *Figure 5—figure supplement 2E*, and *Figure 5—figure supplement 9*).

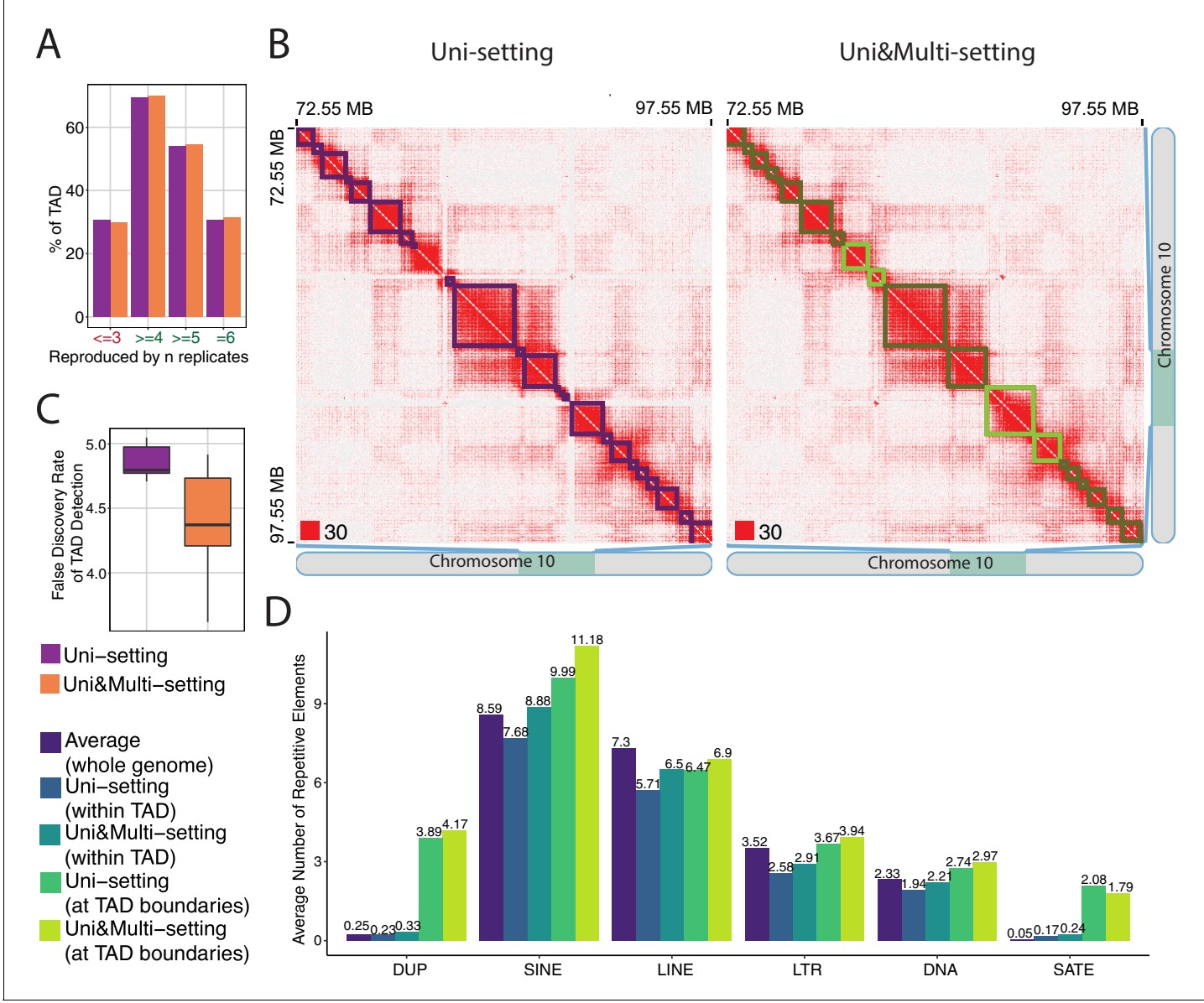

**Figure 5.** mHi-C rescued multi-reads refine detected topologically associating domains. (**A**) Percentage of topologically associating domains (TADs) that are reproducibly detected under Uni-setting and Uni&Multi-setting. TADs that are not detected in at least 4 of the six replicates are considered as non-reproducible. (**B**) Comparison of the contact matrices with superimposed TADs between Uni- and Uni&Multi-setting for chr10:72,550,000– 97,550,000. Red squares at the left bottom of the matrices indicate the color scale. TADs affected by white gaps involving repetitive regions are highlighted in light green. Light green outlined areas correspond to new TAD boundaries. (**C**) False discovery rate of TADs detected under two settings. TADs that are not reproducible and lack CTCF peaks at the TAD boundaries are labeled as false positives. (**D**) Average number of repetitive elements at the boundaries of reproducible TADs compared to those within TADs and genomewide intervals of the same size for GM12878 at 5 kb resolution.

DOI: https://doi.org/10.7554/eLife.38070.052

The following source data and figure supplements are available for figure 5:

**Source data 1.** Topologically associating domains detected by DomainCaller (*Dixon et al., 2012*) under Uni&Multi-setting for six replicates of IMR90.
DOI: https://doi.org/10.7554/eLife.38070.064
**Source data 2.** Topologically associating domains detected by Arrowhead (*Rao et al., 2014*) under Uni&Multi-setting for ten replicates of GM12878.
DOI: https://doi.org/10.7554/eLife.38070.065
**Figure supplement 1.** The number of topologically associating domains (TADs) detected in each chromosome under Uni-setting and Uni&Multi-setting (IMR90).
DOI: https://doi.org/10.7554/eLife.38070.053

*Figure 5 continued on next page*

*Figure 5 continued*

**Figure supplement 2.** Comparison of CTCF peaks at the boundaries of topologically associating domains (TADs) under Uni-setting and Uni&Multi-setting across six replicates of IMR90.

DOI: https://doi.org/10.7554/eLife.38070.054

**Figure supplement 3.** Novel topologically associating domains (TADs) with CTCF peaks at TAD boundaries (IMR90).

DOI: https://doi.org/10.7554/eLife.38070.055

**Figure supplement 4.** Existing topologically associating domains (TADs) with adjusted boundaries supported by CTCF peaks at the new TAD boundaries (IMR90).

DOI: https://doi.org/10.7554/eLife.38070.056

**Figure supplement 5.** Existing topologically associating domains (TADs) with adjusted boundaries supported by CTCF peaks at the new TAD boundaries (IMR90).

DOI: https://doi.org/10.7554/eLife.38070.057

**Figure supplement 6.** False positive topologically associating domains (TADs) detected by the Uni-setting due to the missing reads in low mappability regions (IMR90).

DOI: https://doi.org/10.7554/eLife.38070.058

**Figure supplement 7.** False positive topologically associating domains (TADs) detected by the Uni-setting due to the missing reads in low mappability regions (IMR90).

DOI: https://doi.org/10.7554/eLife.38070.059

**Figure supplement 8.** False positive topologically associating domains (TADs) detected by the Uni-setting due to the missing reads in low mappability regions (IMR90).

DOI: https://doi.org/10.7554/eLife.38070.060

**Figure supplement 9.** False discovery rate of TADs detected under Uni-setting and Uni&Multi-setting (IMR90).

DOI: https://doi.org/10.7554/eLife.38070.061

**Figure supplement 10.** Percentage of TAD boundaries co-localized with different types of repetitive elements under Uni-setting and Uni&Multi-setting for IMR90 at 40 kb and GM12878 at 5 kb.

DOI: https://doi.org/10.7554/eLife.38070.062

**Figure supplement 11.** Average number of repetitive elements at the reproducible topologically associating domains detected under Uni-setting and Uni&Multi-setting for IMR90 at 40 kb.

DOI: https://doi.org/10.7554/eLife.38070.063

---

Next, we assessed the abundance of different classes of repetitive elements, from the Repeat-Masker (*Open R, 2015*) and UCSC genome browser (*Tyner et al., 2017*) hg19 assembly, at the reproducible TAD boundaries. Specifically, we considered segmental duplications (DUP), short interspersed nuclear elements (SINE), long interspersed nuclear elements (LINE), long terminal repeat elements (LTR), DNA transposon (DNA) and satellites (SATE). We utilized ± bin on either side of the edge coordinate of a given domain as its TAD boundary. At a lower resolution, that is 40 kb for IMR90, each boundary is 120 kb region and the percentages of TAD boundaries with each type of repetitive element illustrate negligible differences between the Uni-setting and Uni&Multi-setting (*Figure 5—figure supplement 10A*). Similarly, due to the large sizes of the TAD boundaries, a majority of TAD boundaries harbor SINE, LINE, LTR, and DNA transposon elements. However, higher resolution analysis of the GM12878 dataset at 5 kb reveals SINE elements as the leading category of elements that co-localizes with more than 99% of TAD boundaries followed by LINEs (*Figure 5—figure supplement 10B*). This is consistent with the fact that SINE and LINE elements are relatively short and cover a larger portion of the human genome compared to other subfamilies (15% for SINE and 21% for LINE; *Treangen and Salzberg, 2012*). We further quantified the enrichment of repetitive elements at TAD boundaries by comparing their average abundance with those within TADs and the genomic intervals of the same size across the whole genome as the baseline. *Figure 5D* and *Figure 5—figure supplement 11* show that SINE elements, satellites, and segmental duplications are markedly enriched at the TAD boundaries compared to the whole genome and within TADs. More interestingly, at higher resolution, that is 5 kb for GM12878, the SINE category both have the highest average enrichment and is enhanced by mHi-C (*Figure 5D*). In summary, under Uni&Multi-setting, the detected TAD boundaries tend to harbor more SINE elements supporting prior work that human genome folding is markedly associated with the SINE family (*Cournac et al., 2016*).

## Large-scale evaluation of mHi-C with computational trimming experiments and simulations establishes its accuracy

Before further investigating the accuracy of mHi-C rescued multi-reads with computational experiments, we considered heuristic strategies for rescuing multi-reads at different stages of the Hi-C analysis pipeline as alternatives to mHi-C (*Figure 6A*; see Materials and methods for detailed descriptions of the model-free approaches and related analysis). Specifically, AlignerSelect and DistanceSelect rescue multi-reads by simply choosing one of the alignments of a multi-read pair by default aligner strategy and based on distance, respectively. In addition to these, we designed a direct alternative to mHi-C, named SimpleSelect, as a model-free approach that imposes genomic distance priority in contrast to leveraging of the local interaction signals of the bins by mHi-C (e.g., local contact counts due to other read pairs in candidate bin pairs).

To evaluate the accuracy of mHi-C in a setting with ground truth, we carried out trimming experiments with the A549 151 bp read length dataset and Hi-C data simulations where we compared mHi-C to both a random allocation strategy as the baseline and the additional heuristic approaches we developed (*Figure 6A*). Specifically, we trimmed the set of 151 bp uni-reads from A549 into read lengths of 36 bp, 50 bp, 75 bp, 100 bp, and 125 bp. As a result, a subset of uni-reads at the full read length of 151 bp with known alignment positions were reduced into multi-reads, generating gold-standard multi-read sets with known true origins. The resulting numbers of valid uni- and multi-reads are summarized in *Figure 6—figure supplement 1A* in comparison with the numbers of valid uni-reads in the original A549 datasets. The corresponding multi-to-uni ratios of these settings vary with the lengths of the trimmed reads, and their range covers the typical multi-to-uni ratios observed in the full read length datasets (*Figure 6—figure supplement 1B*).

We first investigated the multi-read allocation accuracy with respect to trimmed read length, sequencing depth, and mappability at resolution 40 kb. *Figure 6B* exhibits superior performance of mHi-C over both the model-free methods and the random baseline in correctly allocating multi-reads of different lengths to their true origins across intra- and inter-chromosomal contacts. As expected and illustrated by *Figure 6B*, the accuracy of multi-read assignment has an increasing trend with the read length. Specifically, it ranges between 70% and 90% for mHi-C and 20% and 35% for the random allocation strategy for the shortest and longest trimmed read lengths of 36 bp and 125 bp, respectively. When the allocated multi-reads are stratified as a function of the mappability, reads with the lowest mappability (<0.1) have accuracy levels of less than 32% to 70% across the trimmed read lengths (*Figure 6C* for 75 bp, *Figure 6—figure supplement 2* for the other trimming lengths). Notably, the accuracy quickly reaches 74% to 87% for reads with mappability of at least 0.5 (*Figure 6C*, *Figure 6—figure supplement 2*).

Next, we assessed the allocation accuracy among different classes of repetitive elements (*Figure 6D*). Allocations involving segmental duplication regions exhibit a systematically lower performance compared to other repeat classes and the overall average across the whole genome for all five trimming settings. Notably, even for these segmental duplication regions, the accuracy of mHi-C is markedly higher than both the model-free approaches and the random selection baseline displayed in *Figure 6B*. To finalize the accuracy investigation, we further varied the trimming setting by mixing uni-reads and multi-reads from different replicates (see setting (ii) of trimming strategies in Materials and methods) and considering resolutions of 10 kb and 40 kb in addition to an empirical Hi-C simulation. *Figure 6—figure supplements 3–6* provide accuracy results closely following the results presented in this section from these additional settings and further validate significantly better performance of mHi-C compared to the random allocation and other heuristic approaches across different trimmed read lengths.

After establishing accuracy, we evaluated the impact of mHi-C rescued multi-reads of the trimmed datasets on the recovery of the (original) full read length contact matrices, topological domain structures, and significant interactions. To assess the recovery of the original contact matrix, we compared both the trimmed Uni- and Uni&Multi-settings with the gold standard Uni-setting at the full read length utilizing HiCRep (*Yang et al., 2017*). *Figure 7A* and *Figure 7—figure supplement 1* illustrate that mHi-C achieves significant improvement in the reproducibility across all chromosomes under all trimming settings compared to the Uni-setting. While the pattern of reproducibility with different read lengths in *Figure 7A* is consistent with the expectation that the longer trimmed reads should yield contact matrices that are more similar to the full read length one,

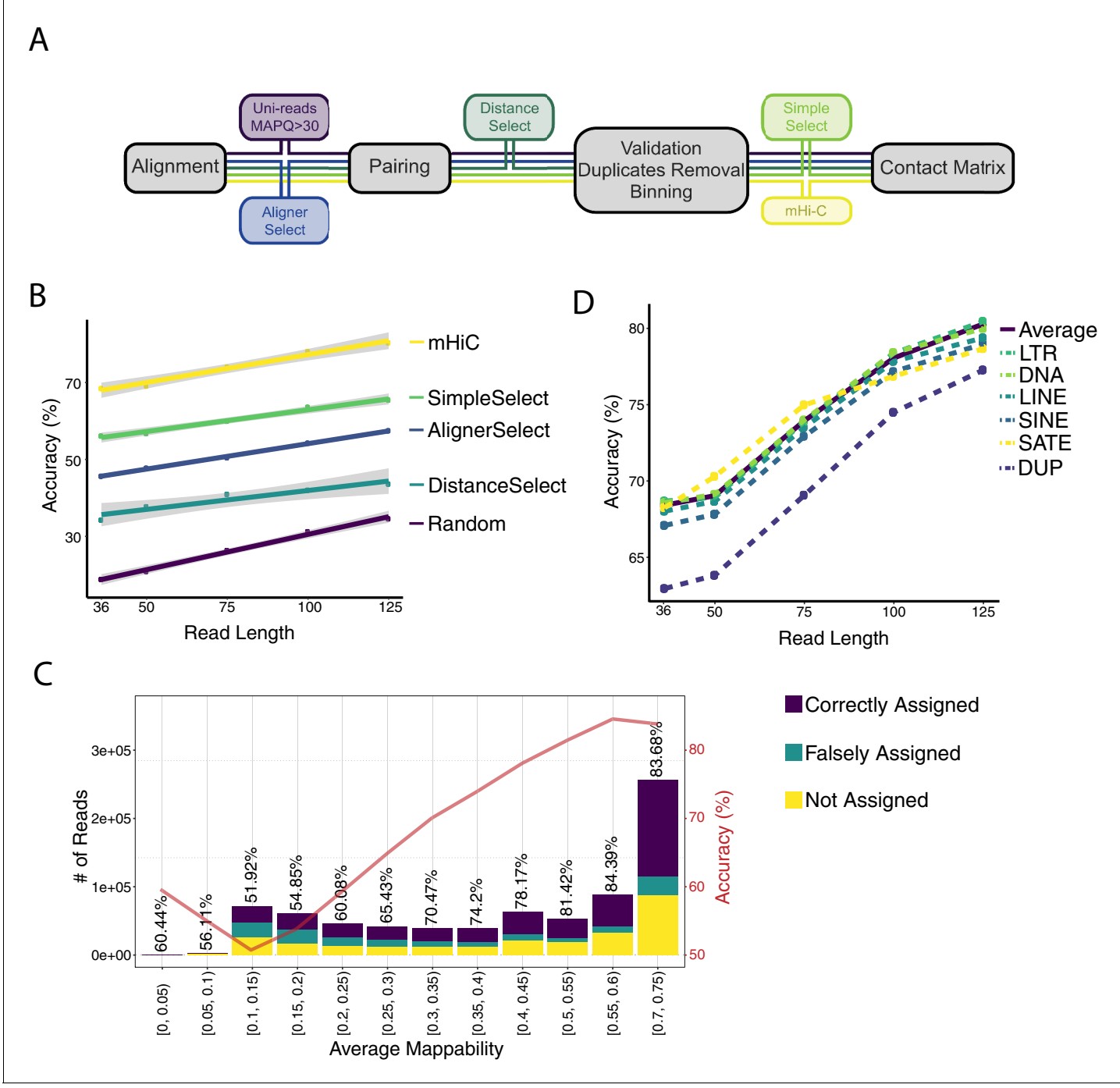

**Figure 6.** Assessing the accuracy of mHi-C allocation by trimming experiments with the A549 study set of 151 bp reads. (**A**) Intuitive heuristic strategies (AlignerSelect, DistanceSelect, SimpleSelect) for model-free assignment of multi-reads at various stages the of Hi-C analysis pipeline. (**B**) Accuracy of mHi-C in allocating trimmed multi-reads with respect to trimmed read length, compared with model-free approaches as well as random selection as a baseline. (**C**) Allocation accuracy with respect to mappability for 75 bp reads. Red solid line depicts the overall accuracy trend. 'Not assigned' category refers to multi-reads with a maximum posterior probability of assignment $\leq 0.5$. (**D**) mHi-C accuracy among different repetitive element classes.

DOI: https://doi.org/10.7554/eLife.38070.066

The following figure supplements are available for figure 6:

**Figure supplement 1.** Summary of the sequencing depths of the full length and trimmed datasets of A549.
DOI: https://doi.org/10.7554/eLife.38070.067

**Figure supplement 2.** Allocation accuracy at the 40 kb resolution among different mappability regions for trimmed reads of varying lengths.
DOI: https://doi.org/10.7554/eLife.38070.068

*Figure 6 continued on next page*

*Figure 6 continued*

**Figure supplement 3.** Intra-chromosomal and intra&inter-chromosomal allocation accuracy with respect to trimmed read length using uni-reads of replicate 1, 3, and 4 combined with multi-reads of replicate 2 (trimming setting (ii)).

DOI: https://doi.org/10.7554/eLife.38070.069

**Figure supplement 4.** Evaluating accuracy of mHi-C allocation with simulations.

DOI: https://doi.org/10.7554/eLife.38070.070

**Figure supplement 5.** Allocation accuracy across different mappability regions for trimmed reads of 36 bp, 50 bp, 75 bp, 100 bp, and 125 bp, using uni-reads of replicate 1, 3, and 4, respectively.

DOI: https://doi.org/10.7554/eLife.38070.071

**Figure supplement 6.** Allocation accuracy across different classes of repetitive elements at 10 kb and 40 kb resolutions using uni-reads of replicates one, three, and combined with multi-reads of replicate two (trimming setting (ii)).

DOI: https://doi.org/10.7554/eLife.38070.072

**Figure supplement 7.** Comparison of significant interactions with respect to genomic distance and life stages between SimpleSelect and mHi-C.

DOI: https://doi.org/10.7554/eLife.38070.073

the improvement in reproducibility due to mHi-C is markedly larger compared to the gains from longer read sequences making multi-read rescue essential. For example, the reproducibility for Uni&Multi-setting at 50 bp is 8.84% to 27.33% higher than that of Uni-setting at 125 bp. We further evaluated reproducibility under each trimming setting across replicates and benchmarked the results against the reproducibility of the Uni-setting with the original read length of 151 bp. *Figure 7—figure supplements 2* and *3* confirm the gain in reproducibility across replicates due to Uni&Multi-setting for all the trimming lengths. As expected, the reproducibility across replicates based on the uni-reads of the original read length is higher than the levels achievable by the Uni&Multi-Setting at trimmed read lengths. This comparison further supports that mHi-C assigns multi-reads in a biologically meaningful manner as was evidenced earlier by the increased reproducibility among replicates of the same condition (*Figure 2B* and *Figure 2—figure supplements 9* and *10*) but not across replicates of the different conditions (*Figure 2—figure supplement 12*). It also rules out the possibility of inflation of the reproducibility metric by consistent but biologically irrelevant assignment of multi-reads to certain loci. TAD identification with these trimmed sets highlights the sensitivity of TAD boundary detection to the sequencing depth. *Figure 7B* and *Figure 7—figure supplements 4–7* display examples where the trimmed Uni&Multi-setting achieved better recovery of TAD structure of the full-length dataset compared to trimmed Uni-setting. Overall, this performance is attributable to the accuracy of mHi-C assignments and the resulting increase in sequencing depth of the trimmed uni-read dataset. Finally, we compared the significant interactions detected by the trimmed Uni- and Uni&Multi-settings and observed that mHi-C rescued multi-reads in trimmed datasets enable detection of a larger number of interactions across a range of FDR thresholds (*Figure 7—figure supplement 8*). Most notably, an evaluation of detection power for the top 10K significant interactions of the full-length dataset demonstrates that, while the Uni-setting can only recover 50% of these at the trimmed read length of 36 bp, Uni&Multi-setting recovers 70% (*Figure 7C* and *Figure 7—figure supplement 9*). We note that these power values are slightly underestimated because the full-length uni-read dataset also included chimeric reads that were rescued as uni-reads. In contrast, trimmed reads in the trimming experiments were generated from uni-reads without rescuing chimeric reads (see Materials and methods; *Figure 7—figure supplement 10*). Despite this, the 26.19% increase in sequencing depth due to multi-reads at the trimmed read length of 36 bp (*Figure 6—figure supplement 1B*) translated into a significantly better recovery of the significant interactions. Further assessment by ROC and PR analysis (*Figure 7D, E* and *Figure 7—figure supplement 11*) of the set of significant contacts identified by both settings illustrates that Uni&Multi-setting exhibits these advantages without inflating the false discoveries. As reads get longer towards the full length, the ROC and PR curves converge under the two settings (*Figure 7—figure supplement 11*).

## Discussion

Hi-C data are powerful for identifying long-range interacting loci, chromatin loops, topologically associating domains (TADs), and A/B compartments (*Lieberman-Aiden et al., 2009*; *Yu and Ren, 2017*). Multi-mapping reads, however, are absent from the typical Hi-C analysis pipelines, resulting

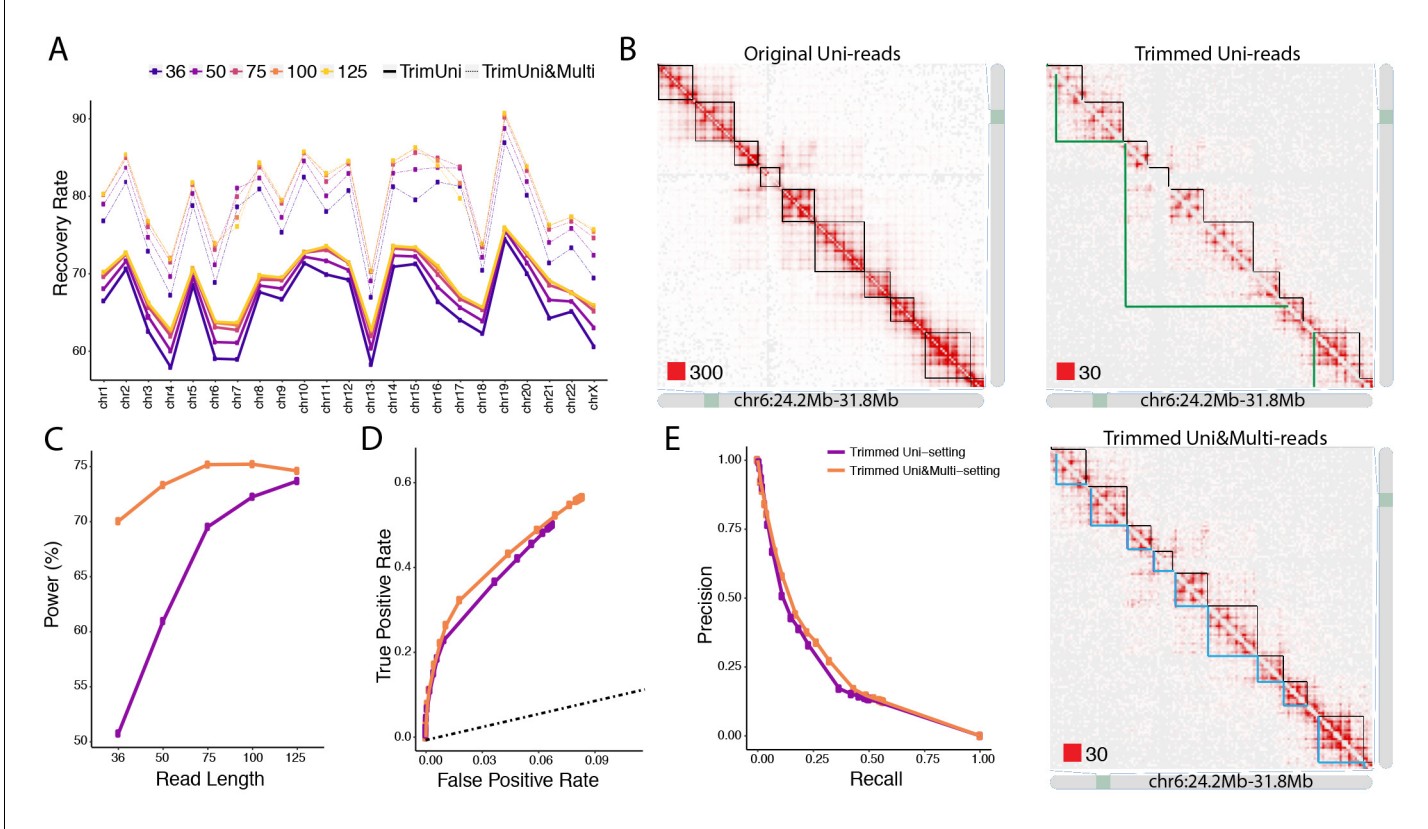

**Figure 7.** Trimmed uni- and multi-reads to recover the original contact matrix of the longer read dataset A549. (**A**) mHi-C rescued multi-reads of the trimmed dataset along with trimmed uni-reads lead to contact matrices that are significantly more similar to original contact matrices compared to only using trimmed uni-reads. (**B**) TAD detection on chromosome six with the original longer uni-reads contact matrix with black TAD boundaries, trimmed uni-reads (36 bp) contact matrix with green TAD boundaries, and trimmed uni- and multi-reads (36 bp) contact matrix with blue TAD boundaries. (**C**) The power of recovering top 10,000 significant interactions of full read length dataset using trimmed reads under FDR10%. (**D, E**) Receiver Operating Characteristic (ROC) and Precision-Recall (PR) curves for trimmed Uni- and Uni&Multi-setting. The ground truth for these curves is based on the significant interactions identified by the full read length dataset at FDR of 10%. The dashed line is y = x.

DOI: https://doi.org/10.7554/eLife.38070.074

The following figure supplements are available for figure 7:

**Figure supplement 1.** Reproducibility of trimmed Uni-setting and trimmed Uni&Multi-setting across different read lengths at 40 kb resolution.
DOI: https://doi.org/10.7554/eLife.38070.075

**Figure supplement 2.** Reproducibility comparison between original Uni-setting and trimmed Uni-setting and Uni&Multi-setting across replicates at 40 kb resolution.
DOI: https://doi.org/10.7554/eLife.38070.076

**Figure supplement 3.** Reproducibility comparison between original Uni-setting and trimmed Uni- and Uni&Multi-settings across chromosomes at 40 kb resolution.
DOI: https://doi.org/10.7554/eLife.38070.077

**Figure supplement 4.** TAD detection on chromosome 3 of original longer uni-reads contact matrix with black TAD boundaries, trimmed uni-reads (36 bp) contact matrix with green TAD boundaries and trimmed uni- and multi-reads (36 bp) contact matrix with blue TAD boundaries.
DOI: https://doi.org/10.7554/eLife.38070.078

**Figure supplement 5.** TAD detection on chromosome 7 of original longer uni-reads contact matrix, trimmed uni-reads (36 bp) contact matrix and trimmed uni- and multi-reads (36 bp) contact matrix.
DOI: https://doi.org/10.7554/eLife.38070.079

**Figure supplement 6.** TAD detection on chromosome 7 of original longer uni-reads contact matrix, trimmed uni-reads (36 bp) contact matrix and trimmed uni- and multi-reads (36 bp) contact matrix.
DOI: https://doi.org/10.7554/eLife.38070.080

**Figure supplement 7.** TAD detection on chromosome 10 of original longer uni-reads contact matrix with black TAD boundaries, trimmed uni-reads (36 bp) contact matrix with green TAD boundaries and trimmed uni- and multi-reads (36 bp) contact matrix with blue TAD bounaries.
DOI: https://doi.org/10.7554/eLife.38070.081

*Figure 7 continued on next page*

*Figure 7 continued*

**Figure supplement 8.** Numbers of significant interactions identified with trimmed reads under Uni- and Uni&Multi-settings at FDR 0.1%, 1%, 5%, 10%.
DOI: https://doi.org/10.7554/eLife.38070.082

**Figure supplement 9.** Power is computed as the percentage of top 10,000 significant interactions of the full read length dataset detected by the analysis of trimmed read datasets under FDR of 10%.
DOI: https://doi.org/10.7554/eLife.38070.083

**Figure supplement 10.** Comparison of read decompositions of 'rep2' with 'rep2-NonChimericReads' sequencing depths indicates that in the full read length dataset, a large proportion of the uni-reads are due to rescued chimeric reads (i.e., these are Uni- or Multi-reads in rep2 and Singletons in rep2-NonChimericReads).
DOI: https://doi.org/10.7554/eLife.38070.084

**Figure supplement 11.** ROC and PR curves for detection of full read length dataset significant interactions by the analysis of trimmed read datasets under the trimmed Uni- and Uni&Multi-settings at read lengths of 50 bp, 75 bp, 100 bp, and 125 bp.
DOI: https://doi.org/10.7554/eLife.38070.085

in under-representation and under-study of three-dimensional genome organization involving repetitive regions. Consequently, downstream analysis of Hi-C data relies on the incomplete Hi-C contact matrices which have frequent and, sometimes, severe interaction gaps spanning across the whole matrix. While centromeric regions contribute to such gaps, our results indicate that lack of multi-reads in the analysis is a significant contributor. Our Hi-C multi-read mapping strategy, mHi-C, probabilistically allocates high-quality multi-reads to their most likely positions (*Figure 1E*) and successfully fills in the missing chunks of contact matrices (*Figure 2A* and *Figure 2—figure supplements 1–4*). As a result, incorporating multi-reads yields remarkable increase in sequencing depth which is translated into significant and consistent gains in reproducibility of the raw contact counts (*Figure 2B*) and detected interactions (*Figure 2C*). Analysis with mHi-C rescued reads identifies novel significant interactions (*Figure 3*), promoter-enhancer interactions (*Figure 4*), and refines domain structures (*Figure 5*). Our computational experiments with trimmed and simulated Hi-C reads validate mHi-C and elucidate the significant impact of multi-reads in all facets of the Hi-C data analysis. We demonstrate that even for the shortest read length of 36 bp, mHi-C accuracy exceeds 74% (85% for longer trimmed reads) for regions with underlying mappability of at least 0.5 (*Figure 6C* and *Figure 6—figure supplement 2*). mHiC significantly outperforms a baseline random allocation strategy as well as several other model-free and intuitive multi-read allocation strategies while achieving its worst allocation accuracy of 63% for reads originating from segmental duplications (*Figure 6B and D*). Trimming experiments further demonstrated the utility of multi-reads for contact matrix, TAD, and significant interaction recovery (*Figure 7*).

The default setting of mHi-C is intentionally conservative. In this default setting, mHi-C rescues high-quality multi-reads that can be allocated to a candidate alignment position with a high probability of at least 0.5. mHi-C allows relaxation of this strict filtering where instead of keeping reads with allocation probability greater than 0.5, these posterior allocation probabilities can be utilized as fractional contacts. We chose not to pursue this approach in this work as the current downstream analysis pipelines do not accommodate such fractional contacts. Currently, mHi-C model does not take into account potential copy number variations and genome arrangements across the genome. While mHi-C model can be extended to take into account estimated copy number and arrangement maps of the underlying sample genomes as we have done for other multi-read problems (*Zhang and Keleş, 2014*), our computational experiments with cancerous human alveolar epithelial cells A549 does not reveal any notable deterioration in mHi-C accuracy for these cells with copy number alternations.

## Materials and methods

### mHi-C workflow

We developed a complete pipeline customized for incorporating high-quality multi-mapping reads into the Hi-C data analysis workflow. The overall pipeline, illustrated in *Figure 1—figure supplements 1* and *2*, incorporates the essential steps of the Hi-C analysis pipelines. In what follows, we outline the major steps of the analysis to explicitly track multi-reads and describe how mHi-C utilizes them.

## Read end alignment: uni- and multi-reads and chimeric reads

The first step in the mHi-C pipeline is the alignment of each read end separately to the reference genome. The default aligner in the mHi-C software is BWA (*Li and Durbin, 2010*); however mHi-C can work with any aligner that outputs multi-reads. The default alignment parameters are (i) edit distance maximum of 2 including mismatches and gap extension; and (ii) a maximum number of gap open of 1. mHi-C sets the maximum number of alternative hits saved in the XA tag to be 99 to keep track of multi-reads. If the number of alternative alignments exceeds the threshold of 99 in the default setting, these alignments are not recorded in XA tag. We regarded these alignments as low-quality multi-mapping reads compared to those multi-mapping reads that have a relatively smaller number of alternative alignments. In summary, low-quality multi-mapping reads are discarded together with unmapped reads, only leaving uniquely mapping reads and high-quality multi-mapping reads for downstream analysis. mHi-C pipeline further restricts the maximum number of mismatches (maximum to be 2 compared to three in BWA default setting) to ensure that the alignment quality of multi-reads is comparable to that of standard Hi-C pipeline.

Chimeric reads, that span ligation junction of the Hi-C fragments (*Figure 1—figure supplement 1*) are also a key component of Hi-C analysis pipelines. The ligation junction sequence can be derived from the restriction enzyme recognition sites and used to rescue chimeric reads. mHi-C adapts the pre-splitting strategy of diffHiC (*Lun and Smyth, 2015*), which is modified from the existing Cutadapt (*Martin, 2011*) software. Specifically, the read ends are trimmed to the center of the junction sequence. If the trimmed left 5' ends are not too short, for example $\geq$ 25 bps, these chimeric reads are remapped to the reference genome. As the lengths of the chimeric reads become shorter, these reads tend to become multi-reads.

## Valid fragment filtering

While each individual read end is aligned to reference genome separately, inferring interacting loci relies on alignment information of paired-ends. Therefore, read ends are paired after unmapped and singleton read pairs as well as low-quality multi-mapping ends (*Figure 1—figure supplement 1* and *Supplementary file 1*) are discarded. After pairing, read end alignments are further evaluated for their representation of valid ligation fragments that originate from biologically meaningful long-range interactions (*Figure 1—figure supplement 1*). First, reads that do not originate from around restriction enzyme digestion sites are eliminated since they primarily arise due to random breakage by sonication (*Belaghzal et al., 2017*). This is typically achieved by filtering the reads based on the total distance of two read end alignments to the restriction site. We required the total distance to be within 50–800 bps for the mammalian datasets and 50–500 bps for *P. falciparum*. The lower bound of 50 for this parameter is motivated by the chimeric reads with as short as 25 bps on both ends. Second, a single Hi-C interaction ought to involve two restriction fragments. Therefore, read ends falling within the same fragment, either due to dangling end or self-circle ligation, are filtered. Third, because the nature of chromatin folding leads to the abundance of random short-range interactions, interactions between two regions that are too close in the genomic distance are highly likely to be random interaction without regulatory implications. As a result, reads with ends aligning too close to each other are also filtered according to the twice the resolution rule. Notably, as a result of this valid fragment filtering, some multi-mapping reads can be counted as uniquely mapping reads (*Supplementary file 1 - 2b*). This is because, although a read pair has multiple potential genomic origins dictated by its multiple alignments, only one of them ends up passing the validation screening. Once the multi-mapping uncertainty is eliminated, such read pairs are passed to the downstream analysis as uni-reads. We remark here that standard Hi-C analysis pipelines do not rescue these multi-reads.

## Duplicate removal

To remove PCR duplicates, mHi-C considers the following two issues. First, due to allowing a maximum number of 2 mismatches in alignment, some multi-reads may have the exact same alignment position and strand direction with uni-reads. If such duplicates arise, uni-reads are granted higher priority and the overlapping multi-reads together with all their potential alignment positions are discarded completely. This ensures that the uni-reads that arise in standard Hi-C analysis pipelines will not be discarded as PCR duplicates in the mHi-C pipeline. Second, if a multi-mapping read

alignment is duplicated with another multi-read, the read with smaller alphabetical read query name will be preserved. More often than not, if multi-read A overlaps multi-read B at a position, then it is highly likely that they will overlap at other positions as well. This convention ensures that it is always the read pair A alignments that are being retained (*Figure 1—figure supplement 2*).

## Genome binning

Due to the typically limited sequencing depths of Hi-C experiments, the reference genome is divided into small non-overlapping intervals, that is bins, to secure enough number of contact counts across units. The unit can be fix-sized genomic intervals or a fixed number of consecutive restriction fragments. mHi-C can switch between the two unit options with ease. After binning, the interaction unit reduces from alignment position pairs to bin pairs. Remarkably, multi-mapping reads, ends of which are located within the same bin pair, reduce to uni-reads as their potential multi-mapping alignment position pairs support the same bin pair contact. Therefore, there is no need to distinguish the candidate alignments within the same bin (*Figure 1—figure supplement 2* and *Supplementary file 1 - 3b*).

## mHi-C generative model and parameter estimation

mHi-C infers genomic origins of multi-reads at the bin pair level (*Supplementary file 1*). We denoted the whole alignment vector for a given paired-end read $i$ by vector $\mathbf{Y}_i$. If the two read ends of read $i$ align to only bin $j$ and bin $k$, respectively, we set the respective components of the alignment vector as: $Y_{i,(j,k)} = 1$ and $Y_{i,(j',k')} = 0, \forall \ j' \neq j, k' \neq k$. Index of read, $i$, ranges from 1 to $N$, where $N$ is total number of valid Hi-C reads, including both uni-reads and multi-reads that pass the essential processing in *Figure 1—figure supplements 1* and *2*. Overall, the reference genome is divided into $M$ bins and $j$ represents the bin index of the end, alignment position of which is upstream compared to the other read end position indicated by $k$. Namely, $j$ takes on a value from 1 to $M-1$ and $k$ runs from $j+1$ to the maximum value $M$. For uniquely mapping reads, only one alignment is observed, that is $\sum_{(j,k)}^{(M-1,M)} Y_{i,(j,k)} = 1$. However, for multi-mapping reads, we have $\sum_{(j,k)}^{(M-1,M)} Y_{i,(j,k)} > 1$.

We next defined a hidden variable $Z_{i,(j,k)}$ to denote the true genomic origin of read $i$. If read $i$ originates from position bin pairs $j$ and $k$, we have $Z_{i,(j,k)} = 1$. In addition, a read can only originate from one alignment position pair on the genome; thus, $\sum_{(j,k)}^{(M-1,M)} Z_{i,(j,k)} = 1$ for both uni- and multi-reads. We define $O_i = \{(j, k): Z_{i,(j,k)} = 1\}$ to represent true genomic origin of read $i$ and $S_{O_i}$ as the set of location pairs that read pair $i$ can align to. Hence, $Y_{i,(j,k)} = 1$, if $(j, k) \in S_{O_i}$. Under the assumption that the true alignment can only originate from those observed alignable positions, $O_i$ must be one of the location pairs in $S_{O_i}$. We further assume that the indicators of true origin for read $i$, $Z_i = (Z_{i,(1,2)}, Z_{i,(1,3)}, ..., Z_{i,(M-1,M)})$ are random draws from a Dirichlet - Multinomial distribution. Specifically,

$$\mathbf{Z_i} \overset{i.i.d.}{\sim} \text{Multinomial}(\boldsymbol{\pi}_{(1,2)}, \boldsymbol{\pi}_{(1,3)}, \cdots, \boldsymbol{\pi}_{(j,k)}, \cdots, \boldsymbol{\pi}_{(M-1,M)}), \quad i = 1, \cdots, N, \tag{1}$$

where $\pi_{(j,k)}$ can be interpreted as contact probability between bin $j$ and $k$ ($j<k$). We further assume that

$$\boldsymbol{\pi} \sim \text{Dirichlet}(\boldsymbol{\gamma}_{(1,2)}, \boldsymbol{\gamma}_{(1,3)}, \cdots, \boldsymbol{\gamma}_{(j,k)}, \cdots, \boldsymbol{\gamma}_{(M-1,M)}), \tag{2}$$

where $\boldsymbol{\pi} = (\pi_{(1,2)}, \pi_{(1,3)}, \cdots, \pi_{(M-1,M)})$ and $\gamma_{(j,k)}$ is a function of genomic distance and quantifies random contact probability. Specifically, we adapt the univariate spline fitting approach from Fit-Hi-C (*Ay et al., 2014a*) for estimating random contact probabilities with respect to genomic distance and set $\gamma_{(j,k)} = Spline(j,k) \times N + 1$. Here, $N$ is the total number of valid reads as defined above and $Spline(j,k)$ denotes the spline estimate of the random contact probability between bins $j$ and $k$. Therefore, $Spline(j,k) \times N$ is the average random contact counts (i.e., pseudo-counts) between bin $j$ and $k$. As a result, the probability density function of $\pi$ can be written as:

$$P(\pi|\gamma) = \frac{\Gamma(\sum_{j=1}^{M-1}\sum_{k=j+1}^{M}\gamma_{(j,k)})}{\prod_{j=1}^{M-1}\prod_{k=j+1}^{M}\Gamma(\gamma_{(j,k)})}\prod_{j=1}^{M-1}\prod_{k=j+1}^{M}\pi_{(j,k)}^{(\gamma_{(j,k)}-1)}$$

$$= \frac{\Gamma(\sum_{j=1}^{M-1}\sum_{k=j+1}^{M}(Spline(j,k)\times N+1))}{\prod_{j=1}^{M-1}\prod_{k=j+1}^{M}\Gamma(Spline(j,k)\times N+1)}\prod_{j=1}^{M-1}\prod_{k=j+1}^{M}\pi_{(j,k)}^{Spline(j,k)\times N}.$$

We next derive the full data joint distribution function.

**Lemma 1**. Given the true genomic origin under the mHi-C setting, the set of location pairs that a read pair can align to will have observed alignments with probability 1.

**Proof**.

$$P(Y_i|Z_{i,(j,k)}=1) = (Y_i|O_i)$$

$$= \prod_{j=1}^{M-1}\prod_{k=j+1}^{M}P(Y_{i,(j,k)}|O_i)$$

$$= \prod_{j=1}^{M-1}\prod_{k=j+1}^{M}[1(Y_{i,(j,k)}=1,(j,k)\in S_{O_i})+1(Y_{i,(j,k)}\neq 1,(j,k)\notin S_{O_i})]$$

$$= 1. \quad \square$$

Based on Lemma 1, we can get the joint distribution $P(\mathbf{Y},\mathbf{Z}|\pi)$ as

$$P(Y,\mathbf{Z}|\pi) = \prod_i^N P_\pi(\mathbf{Y_i},\mathbf{Z_i})$$

$$= \prod_i^N\prod_{j=1}^{M-1}\prod_{k=j+1}^{M}P_\pi(\mathbf{Y_i},Z_{i,(j,k)}=1)^{Z_{i,(j,k)}}$$

$$= \prod_i^N\prod_{j=1}^{M-1}\prod_{k=j+1}^{M}[P_\pi(\mathbf{Y_i}|Z_{i,(j,k)}=1)\pi_{(j,k)}]^{Z_{i,(j,k)}}$$

$$= \prod_i^N\prod_{j=1}^{M-1}\prod_{k=j+1}^{M}\pi_{(j,k)}^{Z_{i,(j,k)}}.$$

Using the Dirichlet-Multinomial conjugacy, we derive the posterior distribution of $\pi$ as

$$P(\pi|\mathbf{Z}) \propto P(\pi,\mathbf{Z}) = \prod_{i=1}^{N}P(\mathbf{Z_i}|\pi)P(\pi)$$

$$\propto \prod_{j=1}^{M-1}\prod_{k=j+1}^{M}\pi_{(j,k)}^{(\sum_{i=1}^{N}Z_{i,(j,k)}+\gamma_{(j,k)}-1)}$$

$$= \prod_{j=1}^{M-1}\prod_{k=j+1}^{M}\pi_{(j,k)}^{(\sum_{i=1}^{N}Z_{i,(j,k)}+Spline(j,k)\times N)}.$$

We next derive an Expectation-Maximization algorithm for fitting this model.

**E-step**.

$$Z_{i,(j,k)}^{(t)} = E(Z_{i,(j,k)}|Y_i,\pi) = \frac{\pi_{(j,k)}^{(t)}}{\sum_{(j',k')\in S_{O_i}}\pi_{(j',k')}^{(t)}}1[(j,k)\in S_{O_i}].$$

**M-step**.

$$\pi_{(j,k)}^{(t+1)} = \frac{\sum_{i=1}^{N}Z_{i,(j,k)}^{(t)}+Spline(j,k)\times N}{N+\sum_{j'=1}^{M-1}\sum_{k'=j+1}^{M}Spline(j',k')\times N}.$$

Estimate of the contact probability $\pi_{(j,k)}$ in the M-step can be viewed as an integration of local interaction signal, encoded in $\sum_{i=1}^{N}Z_{i,(j,k)}^{(t)}$, and random contact signal due to prior, that is, $Spline(j,k)\times N$.

The by-products of the EM algorithm are posterior probabilities, $P(Z_{i,(j,k)}=1 |Y_i, \pi)$, which are utilized for assigning each multi-read to the most likely genomic origin. To keep mHi-C output compatible with the input required for the widely used significant interaction detection methods, we filtered multi-reads with maximum allocation posterior probability less than or equal to 0.5 and assigned the remaining multi-reads to their most likely bin pairs. This ensured the use of at most one bin pair for

each multi-read pair. We repeated our computational evaluations by varying this threshold on the posterior probabilities to ensure robustness of the overall conclusions to this threshold.

## Assessing false positive rates for significant interactions and TADs identification under the Uni- and Uni&Multi-settings

To quantify false positive rates of the Uni- and Uni&Multi-settings at the significant interaction level, we defined true positives and true negatives by leveraging deeply sequenced replicates of the IMR90 dataset (replicates 1–4). Significant interactions reproducibly identified across all four replicates at 0.1% FDR by both the Uni- and Uni&Multi-settings were labeled as true positives (i.e., true interactions). True negatives were defined as all the interactions that were not deemed significant at 25% FDR in any of the four replicates. We then evaluated significant interactions identified by smaller depth replicates 5 and 6 with ROC and PR curves (*Figure 3—figure supplement 13*) by using these sets of true positives and negatives as the gold standard. To quantify false positive rates at the topologically associating domains (TADs) level (*Figure 5C*, *Figure 5—figure supplement 2E*), we utilized TADs that are reproducible in more than three replicates of the IMR90 dataset and/or harbor CTCF peaks at the boundaries as true positives. The rest of the TADs are supported neither by multiple replicates nor by CTCF, hence are regarded as false positives.

## Evaluating reproducibility

Reproducibility in contact matrices was evaluated using HiCRep in the default settings. We further assessed the reproducibility in terms of identified interactions by grouping them into three categories: those only detected under Uni-setting, those unique to Uni&Multi-setting, and those that are detected under both settings. The reproducibility is calculated by overlapping significant interactions between every two replicates and recording the percentage of interactions that are also deemed significant in another replicate (*Figure 2—figure supplement 13*).

## Chromatin states of novel significant interactions

We annotated the novel significant interactions with the 15 states ChromHMM segmentations for IMR90 epigenome (ID E017) from the Roadmap Epigenomics project (*Kundaje et al., 2015*). All six replicates of IMR90 are merged together in calculating the average enrichment of significant interactions among the 15 states (*Figure 3D* and *Figure 3—figure supplement 14A*).

## ChIP-seq analysis

ChIP-seq peak sets for IMR90 cells were obtained from ENCODE portal (https://www.encodeproject.org/) and GEO (*Barrett et al., 2013*). Specifically, we utilized H3K4me1 (ENCSR831JSP), H3K4me3 (ENCSR087PFU), H3K36me3 (ENCSR437ORF), H3K27ac (ENCSR002YRE), H3K27me3 (ENCSR431UUY) and CTCF (ENCSR000EFI) from the ENCODE project and p65 (GSM1055810), p300 (GSM1055812) and PolII (GSM1055822) from GEO (*Barrett et al., 2013*). In addition, raw data files in fastq format were processed by Permseq (*Zeng et al., 2015*) utilizing DNase-seq of IMR90 (ENCODE accession ENCSR477RTP) to incorporate multi-reads and, subsequently, peaks were identified using ENCODE uniform ChIP-seq data processing pipeline (https://www.encodeproject.org/pages/pipelines/#DNA-binding). CTCF motif quantification for topologically associating domains was carried out with FIMO (*Grant et al., 2011*) under the default settings using CTCF motif frequency matrix from JASPAR (*Khan et al., 2017*).

## Promoters with significant interactions

Significant interactions across six replicates of the IMR90 study were annotated with GENCODE V19 (*Harrow et al., 2012*) gene annotations and enhancer regions from ChromHMM. Gene expression calculations utilized RNA-seq quantification results from the ENCODE project with accession number ENCSR424FAZ.

## Marginal Hi-C tracks in *Figure 3—figure supplement 15–17*

Uni-setting and Uni&Multi-setting Hi-C tracks displayed on the UCSC genome browser figures (*Figure 3—figure supplement 15–17*) are obtained by aggregating contact counts of six replicates of IMR90 for each genomic coordinate along the genome.

## Visualization of contact matrices and interactions

We utilized Juicebox (*Durand et al., 2016*), HiGlass (*Kerpedjiev et al., 2017*), and WashU epige-nome browser (*Zhou et al., 2011*) for depicting contact matrices and interactions, respectively, throughout the paper. Normalization of the contact matrices for visualization was carried out by the Knight-Ruiz Matrix Balancing Normalization (*Knight and Ruiz, 2013*) provided by Juicebox (*Durand et al., 2016*).

## Model-free multi-reads allocation strategies

The simplified and intuitive strategies depicted in *Figure 6A* correspond to rescuing multi-reads at different essential stages of the Hi-C analysis pipeline. AlignerSelect relies on the base aligner, for example BWA, to determine the primary alignment of each individual end of a multi-read pair. DistanceSelect enables the distance prior to dominate. It selects the read end alignments closest in the genomic distance as the origins of the multi-read pair and defaults to the primary alignment selected by base aligner for inter-chromosomal multi-reads. Finally, SimpleSelect follows the overall mHi-C pipeline closely by making use of the standard Hi-C validation checking and binning proce-dures. For the reads that align to multiple bins, it selects the bin pair closest in the genomic distance as the allocation of the multi-read pair. Bin-pair allocations for inter-chromosomal multi-reads are set randomly in this strategy.

## Comparison of mHi-C with model-free multi-reads allocation for their impact on identifying differential interactions

We evaluated the direct biological consequence of heavily biasing read assignment by genomic dis-tance, as employed by SimpleSelect, by comparing the significant interactions among three life stages of *P. falciparum*. We reasoned that the better multi-reads allocation strategy would reveal a differential analysis pattern more consistent with the Uni-setting, whereas a genomic distance biased strategy - SimpleSelect - will underestimate differences since multi-reads will be more likely to be allocated to candidate bin pairs with shortest genomic distance regardless of other local contact sig-nals (*Figure 6—figure supplement 7A*). *Figure 6—figure supplement 7B* corroborates this draw-back of SimpleSelect and demonstrates that mHi-C differential patterns agree better with that of the Uni-setting. Moreover, *Figure 6—figure supplement 7B* suggests that rings stage is more similar to schizonts, an observation consistent with existing findings on *P. falciparum* life stages (*Ay et al., 2014b*; *Bunnik et al., 2018*).

## Trimming procedures

We considered two approaches for generating evaluation datasets where we combined the trimmed multi-reads from replicate two, which has the median sequencing depth among all replicates of the A549 study set, with (i) trimmed reads of replicate two that remain uniquely aligned to the reference genome at the same trimmed read length (*Figures 6* and *7*), and (ii) uni-reads from other replicates, that is, replicates one, three, and four in the A549 dataset individually (*Figure 6—figure supple-ments 3*, *5* and *6*). The first setting enables a direct comparison of the set of uni- and multi-reads at trimmed read length compared to uni-reads at the full read length to evaluate accuracy. The num-bers of reads are summarized in *Figure 6—figure supplement 1A* along with multi-to-uni ratios in *Figure 6—figure supplement 1B*. In the second trimming setting (ii), the uni-read sets are of the original sequencing depth and the added multi-reads constitute a smaller proportion compared to observed levels in the data (*Figure 6—figure supplement 1C*) due to the chimeric read rescue that was part of full-length datasets (*Figure 7—figure supplement 10*). Therefore, for this setting, we leverage the higher overall depth of the datasets and evaluate the multi-read assignment accuracy at different resolutions, that is, 10 kb and 40 kb.

## Simulation procedures

We devised a simulation strategy that utilizes parameters learned from the Hi-C data and results in data with a similar signal to noise characteristics as the actual data.

1. *Construction of the interaction prior based on the uni-reads fragment interaction frequency list of GM12878 dataset (replicate 6). The frequency list from the prior encompasses both the*

genomic distance effect and local interaction signal strength and forms the basis for simulating restriction fragment interactions.

2. *Generating the restriction enzyme cutting sites for each simulated fragment pair.* After sampling interacting fragments using the frequency list from Step 1, a genomic coordinate within ± bp of the restriction enzyme cutting site and a strand direction are selected randomly. Reads of different lengths (36 bp, 50 bp, 75 bp, 100 bp) are generated starting from these cutting sites.

3. *Mutating the resulting reads.* Mutation and gap rates are empirically estimated based on the aligned uni-reads of replicate 6. The reads from Step two are uniformly mutated with these rates allowing up to 2 mutations and one gap.

4. *Simulate sequence quality scores of the reads.* We utilize the empirical estimation of the distribution regarding the sequence base quality scores across individual locations of the read length and simulate for each read its sequence quality scores at the nucleotide level.

5. *Alignment to the reference genome.* The simulated reads are aligned to the reference genome and filtered for validation as we outline in the mHi-C pipeline, resulting in the set of multi-reads that are utilized by mHi-C.

We generated numbers of multi-reads comparable to those of replicate six. In the final step of the simulation studies, we merged the simulated set of multi-reads with uni-reads of replicates three and six and ran mHi-C step4 (binning) - step5 (prior already available) - step6 (assign multi-reads posterior probability) independently at resolutions 10 kb and 40 kb.

## Software availability

mHi-C pipeline is implemented in Python and accelerated by C. The source codes and instructions for running mHi-C are publicly available at https://github.com/keleslab/mHiC (*Zheng and Keleş, 2019*; copy archived at https://github.com/elifesciences-publications/mHiC). Each step is organized into an independent script with flexible user-defined parameters and implementation options. Therefore, analysis can be carried out from any step of the work-flow and easily fits in high-performance computing environments for parallel computations.

## Acknowledgements

This work was supported by NIH HG009744 and NIH HG007019 (SK). FA is partially supported by Institute Leadership Funds from La Jolla Institute for Allergy and Immunology. We thank Peigen Zhou from the University of Wisconsin–Madison for the insightful discussions on accelerating the pipeline. We also thank the peer reviewers and the Reviewing and Senior *eLife* Editors of this work for their constructive comments.

## Additional information

### Funding

| Funder | Grant reference number | Author |
| --- | --- | --- |
| National Human Genome Research Institute | HG009744 | Sunduz Keles |
| La Jolla Institute for Allergy and Immunology | Institute Leadership Funds | Ferhat Ay |
| National Human Genome Research Institute | HG007019 | Sunduz Keles |

The funders had no role in study design, data collection and interpretation, or the decision to submit the work for publication.

### Author contributions

Ye Zheng, Sunduz Keles, Conceptualization, Resources, Data curation, Software, Formal analysis, Supervision, Funding acquisition, Validation, Investigation, Visualization, Methodology, Writing—original draft, Project administration, Writing—review and editing; Ferhat Ay, Conceptualization, Resources, Data curation, Software, Investigation, Writing—review and editing

## Author ORCIDs

Ye Zheng (iD) http://orcid.org/0000-0002-8806-2761
Sunduz Keles (iD) https://orcid.org/0000-0001-9048-0922

## Decision letter and Author response

Decision letter https://doi.org/10.7554/eLife.38070.102
Author response https://doi.org/10.7554/eLife.38070.103

# Additional files

## Supplementary files

• Supplementary file 1. Hi-C and mHi-C terminology.
DOI: https://doi.org/10.7554/eLife.38070.086

• Transparent reporting form
DOI: https://doi.org/10.7554/eLife.38070.087

## Data availability

GEO and ENCODE accession codes for all the data analyzed in this manuscript are provided in the manuscript. Source data files have been provided for Figures 1, 3, 4, and 5 (some via Dryad http://dx.doi.org/10.5061/dryad.v7k3140). The mHiC software is made available on github https://github.com/keleslab/mHiC (copy archived at https://github.com/elifesciences-publications/mHiC) with proper documentation.

The following dataset was generated:

| Author(s) | Year | Dataset title | Dataset URL | Database and Identifier |
|---|---|---|---|---|
| Zheng Y, Ay F | 2018 | Data from: Generative Modeling of Multi-mapping Reads with mHi-C Advances Analysis of Hi-C Studies | http://dx.doi.org/10.5061/dryad.v7k3140 | Dryad Digital Repository, 10.5061/dryad.v7k3140 |

The following previously published datasets were used:

| Author(s) | Year | Dataset title | Dataset URL | Database and Identifier |
|---|---|---|---|---|
| Jin F, Li Y, Dixon JR, Selvaraj S, Ye Z, Lee AY, Yen CA, Schmitt AD, Espinoza C, Ren B | 2013 | IMR90 Hi-C Dataset | https://www.ncbi.nlm.nih.gov/geo/query/acc.cgi?acc=GSE43070 | NCBI Gene Expression Omnibus, GSE43070 |
| Ay F, Bunnik EM, Varoquaux N, Bol SM, Prudhomme J, Vert JP, Noble WS, Le Roch KG | 2014 | Plasmodium Hi-C Dataset | https://www.ncbi.nlm.nih.gov/geo/query/acc.cgi?acc=GSE50199 | NCBI Gene Expression Omnibus, GSE50199 |
| Rao SSP, Huntley MH, Durand NC, Stamenova EK, Bochkov ID, Robinson JT, Sanborn AL, Machol I, Omer AD, Lander ES, Aiden EL A | 2014 | GM12878 Hi-C Dataset | https://www.ncbi.nlm.nih.gov/geo/query/acc.cgi?acc=GSE63525 | NCBI Gene Expression Omnibus, GSE63525 |
| Dixon JR, Selvaraj S, Yue F, Kim A, Li Y, Shen Y, Hu M, Liu JS, Ren B | 2012 | ESC(2012) Hi-C Dataset | https://www.ncbi.nlm.nih.gov/geo/query/acc.cgi?acc=GSE35156 | NCBI Gene Expression Omnibus, GSE35156 |
| Dixon JR, Xu J, Dileep V, Zhan Y, Song F, Le VT, | 2018 | A549 Hi-C Dataset | https://www.ncbi.nlm.nih.gov/geo/query/acc.cgi?acc=GSE92819 | NCBI Gene Expression Omnibus, GSE92819 |

| Galip Gurkan Yard-ımcı AC, Bann DV, Wang Y, Clark R, Zhang L, Yang H, Liu T, Iyyanki S, An L, Pool C, Sasaki T, Rivera-Mulia JC | | | | |
|---|---|---|---|---|
| Bonev B, Cohen NM, Szabo Q | 2017 | ESC(2017) & Cortex Hi-C Datasets | https://www.ncbi.nlm.nih.gov/geo/query/acc.cgi?acc=GSE96107 | NCBI Gene Expression Omnibus, GSE96107 |

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
