## [Decision Letter]

Thank you for submitting your article "Statistical Methods for Profiling 3-Dimensional Chromatin Interactions from Repetitive Regions of Genomes" for consideration by *eLife*. Your article has been reviewed by three peer reviewers, and the evaluation has been overseen by a Reviewing Editor and a Senior Editor. The following individual involved in review of your submission has agreed to reveal his identity: Mikhail Spivakov (Reviewer #1).

The reviewers have discussed the reviews with one another and the Reviewing Editor has drafted this decision to help you prepare a revised submission.

Summary:

Zheng et al. describe a new approach and associated software tool (mHi-C) that attempts to rescue multi-mappers from Hi-C data. Multi-mappers are normally discarded in most analysis pipelines, but including them may help improve analysis results, particularly at and near repetitive regions of the genome. mHi-C uses a probabilistic model to assign likely multi-read alignment locations based on the expected properties of Hi-C read-pairs, including their average interaction profile as a function of distance between regions along the genome. The authors evaluate the impact mHi-C has on a variety of Hi-C analysis results, including the identification of significant contacts, analysis reproducibility, and TAD boundary identification, ultimately concluding that mHi-C identifies additional chromatin interaction features and generally improves the accuracy and interpretation of Hi-C data.

This work is very timely as there are few (if any) suitable methods available that address multi-mapping reads from Hi-C experiments. mHi-C has the potential to make Hi-C data analysis more complete by avoiding discarding data and could potentially enable the analysis of how repetitive regions interact in the nucleus, which until this point has been largely ignored.

While the authors' approach appears reasonable, the main limitation of the manuscript is that the improvement in Hi-C analysis enabled by using mHi-C to re-interpret multi-mappers is difficult to judge, even for someone with experience looking at Hi-C data. More specifically, the primary concern is that the analysis lacks a gold standard and negative control datasets to properly interpret the reported improvement in Hi-C analysis afforded by mHi-C.

Essential revisions:

1) In an effort to improve the interpretation of mHi-C's impact on Hi-C analysis results, the authors should take a different approach for the evaluation of their method that would improve the manuscript. For example:

- It would be good to have a clearer idea of the method's accuracy at different ranges of read mappability – ideally, on simulated data derived using methods such as Sim3C (https://doi.org/10.1093/gigascience/gix103).

- The authors use a single mammalian dataset for validation, but it would be useful to evaluate its accuracy and gain in interaction detection power for a range of mammalian datasets of different quality (as assessed, for example, by cis/trans-ratios) and coverage, generated by different labs.

- Take a set of publicly available long read Hi-C experiments and artificially truncate the sequencing reads to create a gold-standard data set that could be used to evaluate how well mHi-C "recovers" the true alignment positions using the truncated reads. The truncated data, the original data, and the inferred mHi-C analysis of the truncated data could then be compared to evaluate how accurately mHi-C recovered the original read alignment positions, and then the improvement of Hi-C analysis metrics could be compared to see how mHi-C does compared to the true positive result of comparing the original read alignments vs. the truncated set. This would provide the needed context to evaluate how these metrics and analysis scores should be interpreted with respect to a true positive. It would be even better to include a negative control for comparison based on the random assignment of multi-mapper positions using a uniform prior (i.e. randomly take one of the valid read-pair alignments without considering distance etc.).

With these datasets you can estimate how much better the TADs should look if multi-mapper assignment was perfect, and how many more significant interactions there should be, etc., by comparing the true positive data to the truncated data. This analysis has the added benefit of enabling a direct calculation of mHi-C assignment accuracy and could identify which types of multi-mappers are more problematic, and may provide evidence of which repetitive features (i.e. LINE/SINE/segmental duplications/centromeres/etc.) are easy or hard to assign multi-mappers to.

2) Comparison with normalized Hi-C matrix: the authors need to compare their results with either ICE normalized or model-based method normalized Hi-C matrix. As mappability is one of the most important factors during normalization procedure, the results of comparison can be valuable to demonstrate the significance of this work.

3) Figure 5: it would be much more convincing if the authors can further zoom in, and clearly show a few examples where a specific enhancer-promoter interaction is captured only by mHi-C? Maybe virtual 4C is the best way for this purpose.

4) What is the purpose of Figure 3? Why is there a need to introduce an alternative approach to mHi-C model and make it into a main figure?

5) The title suggests that the authors will explore the chromatin interactions from repetitive regions. However, not much analysis was provided on this perspective. It would be really interesting if the authors could delve into this topic. For example, whether certain repetitive elements or sub-class of repetitive elements are involved with the formation of chromatin looping or TADs.

[Editors' note: further revisions were requested prior to acceptance, as described below.]

Thank you for resubmitting your work entitled "Generative Modeling of Multi-mapping Reads with mHi-C Advances Analysis of Hi-C Studies" for further consideration at *eLife*. Your revised article has been favorably evaluated by a Senior Editor, a Reviewing Editor, and three reviewers.

The manuscript has been improved but there are some remaining issues that need to be addressed before acceptance, as outlined below:

One primary concern is "why including multi-mappers would increase the reproducibility of Hi-C contact maps." This might be related to how the concept of 'reproducibility' (Figure 2 and other panels, subsection “Probabilistic assignment of multi-reads results in more complete contact matrices and significantly improves reproducibility across replicates”) is defined. It would be important to clearly establish how the improvement in reproducibility by multi-mapper is not a consequence of artifacts. There are other concerns with the TAD analysis, and unsatisfactory responses to two issues raised by one reviewer (#3).

*Reviewer #1:*

I am happy with the changes made and recommend accepting the paper for publication.

*Reviewer #2:*

The authors have considerably improved the manuscript, implementing many of the suggestions from the previous review and refining their results.

Concerns on the revision:

The primary concern in the revision is how the authors discuss the concept of 'reproducibility' (Figure 2 and other panels, subsection “Probabilistic assignment of multi-reads results in more complete contact matrices and significantly improves reproducibility across replicates”). Specifically, it is not clear conceptually why including multi-mappers would increase the reproducibility of Hi-C contact maps. It is clear that proper inclusion of multi-mappers should increase the completeness of the data, but since there is still a decent chance of mis-assignment when assigning multi-mappers (~25% based on Figure 6B), it's hard to understand how the 'reproducibility' should increase with the inclusion of multi-mappers. One would expect the uni-mappers to be the most 'reproducible' part of the data since they can be accurately placed, so adding data with 'less reproducible' mapping rates to the experiment should, if anything, reduce the actual reproducibility.

One (likely) explanation for this is that mHi-C assigns multi-mappers 'reproducibly' to the same loci across multiple experiments (even if these assignments are not appropriate), which might give the illusion of higher reproducibility. An important control for this is to compare the computationally trimmed dataset reproducibility to the original uni-read reproducibility, which is unfortunately omitted in Figure 7A, which only compares trimmed uni-mappers to trimmed uni+multi mappers. If the original uni-reads are in fact less reproducible than the trimmed-uniq+multi mapper experiments, there is a very good chance the reproducibility metric is being inflated by consistent assignment of multimappers to certain loci by mHi-C. This is not necessarily a major problem with the method (since the authors demonstrate nicely that mHi-C provides many benefits for Hi-C analysis), but it is a major problem with the interpretation of mHi-C's benefits with respect to reproducibility, which is stated many times throughout the manuscript.

The TAD analysis is still not very convincing (Figure 7B). The contact matrices look really similar between all conditions, but the called TADs for the Trimmed Uni-reads looks different for some reason. For example, the big TAD on the upper left doesn't seem to adhere to the likely TAD in the contact map. This may be a quirk of the TAD identification algorithm, but it doesn't make for a convincing visual case. The contact map intensity for many of the figures is getting clipped at local interactions (i.e. the region near the diagonal is essentially the same color of red), making it hard to see where the key differences are that drive the difference in the TAD calling. I would highly recommend either zooming in on smaller regions and/or changing the color scheme/scaling so that the TAD calls and the contact map can be more clearly evaluated by the reader.

*Reviewer #3:*

The authors have addressed most of my concerns. Two questions remain:

1) Figure 4A: I am confused. First of all, the second track shouldn't be there – that's the whole chromosome. I think what the authors intended to plotted is the region in the tiny blue box. Second comment about this figure: if you look at the first track (virtual 4C), in the middle, there is a strong Uni&Multi peak, but when you look at the bottom two tracks (arcs), it shows it is uni-specific interaction. Can author clarify what happened?

2) Regarding this response about its performance on normalized Hi-C matrix, the authors commented that "We updated all the visualizations to display both the normalized and unnormalized versions (Figure 2A, Figure 2—f – —figure supplements 1-4). While the normalized Uni-setting matrices look less incomplete highlighting the impact of normalization, mHi-C generated matrices look visually more complete and display the impact of multi-reads. We provided several such examples as supplementary figures."

I found the answer is not satisfactory – the authors need to provide quantitative evaluation of this claim.

---

## [Author Response]

Essential revisions:1) In an effort to improve the interpretation of mHi-C's impact on Hi-C analysis results, the authors should take a different approach for the evaluation of their method that would improve the manuscript. For example:- It would be good to have a clearer idea of the method's accuracy at different ranges of read mappability – ideally, on simulated data derived using methods such as Sim3C (https://doi.org/10.1093/gigascience/gix103).

We have addressed this point both with trimming experiments where we trim long read datasets to generate shorter read sets and also with simulations where we simulate Hi-C multi-reads. The results displayed in Figures 6B, C and Figure 6—figure supplements 2-5 demonstrate mHi-C’s accuracy in finding correct assignments for multi reads.

These results are incorporated into the subsection “Large-scale evaluation of mHi-C with computational trimming experiments and simulations establishes its accuracy” as follows:

“We first investigated the multi-read allocation accuracy with respect to trimmed read length, sequencing depth, and mappability at resolution 40 kb. […] Figure 6—figure supplements 3-6 provide accuracy results closely following the results presented in this section from these additional settings and further validate significantly better performance of mHi-C compared to the random allocation and other heuristic approaches across different trimmed read lengths.”

- The authors use a single mammalian dataset for validation, but it would be useful to evaluate its accuracy and gain in interaction detection power for a range of mammalian datasets of different quality (as assessed, for example, by cis/trans-ratios) and coverage, generated by different labs.

We have now included 8 datasets from 6 different studies, totaling 4 human, 3 mice, and a *Plasmodium falciparum* dataset with varying characteristics and showed that our main conclusions are consistent across the board. The results pertaining abundance of multi-reads (Figure 1B, D and Figure 1—figure supplements 3-6), their impact on reproducibility (Figure 2B, C and Figure 2—figure supplements 6-9), and gain in power (Figure 3A, B and Figure 3—figure supplements 1-7) are updated with results across all datasets. We further utilized IMR90 and GM12878 datasets along with long read A459 dataset for more detailed analysis, trimming experiments, and simulations. Discussion on the datasets selected is included into the subsection “Multi-reads significantly increase the sequencing depths of Hi-C data” as follows:

“For developing mHi-C and studying its operating characteristics, we utilized six published studies, resulting in eight datasets with multiple replicates, as summarized in Table 1 with more details in Figure 1—source data 1: Table 1. […] Specifically, they span a wide range of sequencing depths (Figure 1B), coverages and *cis*-to-*trans* ratios (Figure 1—figure supplement 3), and have different proportions of mappable and valid reads (Figure 1—figure supplement 4).”

- Take a set of publicly available long read Hi-C experiments and artificially truncate the sequencing reads to create a gold-standard data set that could be used to evaluate how well mHi-C "recovers" the true alignment positions using the truncated reads. The truncated data, the original data, and the inferred mHi-C analysis of the truncated data could then be compared to evaluate how accurately mHi-C recovered the original read alignment positions, and then the improvement of Hi-C analysis metrics could be compared to see how mHi-C does compared to the true positive result of comparing the original read alignments vs. the truncated set. This would provide the needed context to evaluate how these metrics and analysis scores should be interpreted with respect to a true positive. It would be even better to include a negative control for comparison based on the random assignment of multi-mapper positions using a uniform prior (i.e. randomly take one of the valid read-pair alignments without considering distance etc.).With these datasets you can estimate how much better the TADs should look if multi-mapper assignment was perfect, and how many more significant interactions there should be, etc., by comparing the true positive data to the truncated data. This analysis has the added benefit of enabling a direct calculation of mHi-C assignment accuracy and could identify which types of multi-mappers are more problematic, and may provide evidence of which repetitive features (i.e. LINE/SINE/segmental duplications/centromeres/etc.) are easy or hard to assign multi-mappers to.

We would like to thank the reviewers for these excellent suggestions. We carried out the trimming experiment utilizing the long read (151bp) A549 dataset. Specifically, we leveraged this dataset to evaluate mHi-C in terms of:

i) multi-read allocation accuracy, both overall and stratified with respect to read length, mappability, and repetitive elements. Such accuracy is compared with model-free approaches as well as random selection as a baseline. (Figure 6 and Figure 6—figure supplements 2-6);

ii) the ability to recover contact matrix of the full read length dataset that the trimming is based on (Figure 7A, Figure 7—figure supplement 1), ability to recover TAD calls of the full read length dataset (Figure 7B, Figure 7—figure supplements 2-5) and ability to detect significant interactions that can be detected by the full read length dataset (Figure 7C-E and Figure 7—figure supplements 6-8).

The subsection “Large-scale evaluation of mHi-C with computational trimming experiments and simulations establishes its accuracy” is largely dedicated to the results from these trimming experiments.

2) Comparison with normalized Hi-C matrix: the authors need to compare their results with either ICE normalized or model-based method normalized Hi-C matrix. As mappability is one of the most important factors during normalization procedure, the results of comparison can be valuable to demonstrate the significance of this work.

We updated all the visualizations to display both the normalized and unnormalized versions (Figure 2A, Figure 2—figure supplements 1-4). While the normalized Uni-setting matrices look less incomplete highlighting the impact of normalization, mHi-C generated matrices look visually more complete and display the impact of multi-reads. We provided several such examples as supplementary figures.

3) Figure 5: it would be much more convincing if the authors can further zoom in, and clearly show a few examples where a specific enhancer-promoter interaction is captured only by mHi-C? Maybe virtual 4C is the best way for this purpose.

Figure 4A and Figure 4—figure supplements 1-2 now provide higher resolution examples of reproducibly detected novel promoter-enhancer interactions with a 4C like display.

4) What is the purpose of Figure 3? Why is there a need to introduce an alternative approach to mHi-C model and make it into a main figure?

Great question! When we discuss multi-reads in various research communities, everyone seems to come up with various suggestions that have not been applied or implemented yet. Nonetheless, we wanted to summarize the reasonable, albeit heuristic, approaches of allocating multi-reads and compare these with mHi-C. We have now included this figure as supplement (Figure 6—figure supplement 7) and reorganized the related analysis in the subsection “Comparison of mHi-C with model-free multi-reads allocation for their impact on identifying differential interactions”. More importantly, we were able to evaluate these heuristic approaches along with a baseline method of randomly selecting one of the alignments in our trimming experiments and simulations (Figure 6 and Figure 6—figure supplements 3, 4).

5) The title suggests that the authors will explore the chromatin interactions from repetitive regions. However, not much analysis was provided on this perspective. It would be really interesting if the authors could delve into this topic. For example, whether certain repetitive elements or sub-class of repetitive elements are involved with the formation of chromatin looping or TADs.

We completely agree that delving more into this topic would be very interesting. For this manuscript, we evaluated the abundance of repetitive elements at TAD boundaries. This analysis indicated that at lower resolutions (e.g., 40 kb bin size), there are not significant differences in the abundance of repetitive elements under the Uni and Uni&Multi-settings. However, the analysis of GM12878 datasets at the 5 kb resolution clearly illustrates enrichment of the SINE elements, segmental duplications, and satellite repeats at the TAD boundaries compared to within boundaries and intervals of the same size genome-wide intervals (Figure 5D). More interestingly, the SINE category both has the highest average enrichment and is enhanced by mHi-C (Figure 5—figure supplement 11B).

This analysis is now incorporated into the subsection *“*Multi-reads refine the boundaries of topologically associating domains” as follows.

“Next, we assessed the abundance of different classes of repetitive elements, from the RepeatMasker (Open, 2015) and UCSC genome browser (Casper et al., 2017) hg19 assembly, at the reproducible TAD boundaries. […] In summary, under Uni&Multi-setting, the detected TAD boundaries tend to harbor more SINE elements supporting prior work that human genome folding is markedly associated with the SINE family Cournac et al. (2015).”

We also thought that the title “Generative Modeling of Multi-mapping Reads with mHi-C Advances Analysis of High Throughput Genome-wide Conformation Capture Studies” is now more reflective of the research presented in this paper.

[Editors' note: further revisions were requested prior to acceptance, as described below.]

Reviewer #2:The authors have considerably improved the manuscript, implementing many of the suggestions from the previous review and refining their results.Concerns on the revision:The primary concern in the revision is how the authors discuss the concept of 'reproducibility' (Figure 2 and other panels, subsection “Probabilistic assignment of multi-reads results in more complete contact matrices and significantly improves reproducibility across replicates”). Specifically, it is not clear conceptually why including multi-mappers would increase the reproducibility of Hi-C contact maps. It is clear that proper inclusion of multi-mappers should increase the completeness of the data, but since there is still a decent chance of mis-assignment when assigning multi-mappers (~25% based on Figure 6B), it's hard to understand how the 'reproducibility' should increase with the inclusion of multi-mappers. One would expect the uni-mappers to be the most 'reproducible' part of the data since they can be accurately placed, so adding data with 'less reproducible' mapping rates to the experiment should, if anything, reduce the actual reproducibility.One (likely) explanation for this is that mHi-C assigns multi-mappers 'reproducibly' to the same loci across multiple experiments (even if these assignments are not appropriate), which might give the illusion of higher reproducibility. An important control for this is to compare the computationally trimmed dataset reproducibility to the original uni-read reproducibility, which is unfortunately omitted in Figure 7A, which only compares trimmed uni-mappers to trimmed uni+multi mappers. If the original uni-reads are in fact less reproducible than the trimmed-uniq+multi mapper experiments, there is a very good chance the reproducibility metric is being inflated by consistent assignment of multimappers to certain loci by mHi-C. This is not necessarily a major problem with the method (since the authors demonstrate nicely that mHi-C provides many benefits for Hi-C analysis), but it is a major problem with the interpretation of mHi-C's benefits with respect to reproducibility, which is stated many times throughout the manuscript.

We would like to emphasize that the improvement in reproducibility is due to increase in read depth by correctly assigned multi-reads. That said, we welcomed the opportunity to investigate whether there is a systematic bias in multi-read assignment. We looked at this tissue from two angles. First, if there is a systematic bias in multi-read assignment, one might expect the reproducibility of unrelated samples to increase by incorporation of multi-reads. In Figure 2B and Figure 2—figure supplements 9 and 10, we showed that mHi-C leads to significant gains in reproducibility for IMR90 and GM12878. In contrast, Figure 2—figure supplement 12 presents the results from a pairwise replicate reproducibility analysis between 4 replicates of GM12878 and 6 replicates of IMR90 and shows that incorporation of multi-reads do not lead to any significant gain in reproducibility for replicates from unrelated samples (IMR90 vs GM12878) (all Wilcoxon rank-sum test p-values of the pairwise comparisons between Uni- and Uni&Multi-settings > 0.2).

Next, we performed the calculations suggested by the reviewer in the trimming experiments. Figure 7—figure supplements 2 and 3 show that reproducibility across the four replicates of A549 based on the uni-reads of the original read length is higher than the levels achievable by the Uni&Multi-Setting at trimmed read lengths. These two points are incorporated in the text.

In addition, we examined all the incorrectly allocated multi-reads in trimming experiments to explicitly assess whether these multi-reads were consistently being assigned to the same incorrect locations across replicates regardless of the replicate experiment-specific biological signal. The trimmed datasets have uni-reads of replicates 1 to 4 combined with multi-reads resulting from trimming of replicate 2 uni-reads. In each replicate dataset, mHi-C ends up either assigning each multi-read to its true originating location or to an incorrect location. Unless assigned to the true origin, each multi-read can potentially be misassigned to up to 4 different positions (i.e., one wrong location per replicate). Author response image 1 displays the distribution of the numbers of misassigned multi-reads assigned to exactly 1, 2, 3, and 4 locations. We observe that a large portion of the multi-reads are assigned to the same location across the four replicates – while this proportion is large, it is still only 4.78% to 7.79% of all the multi-reads. To evaluate whether these incorrect assignments are nonetheless consistent with the available replicate-specific biological signal, we asked whether their assigned positions corresponded to the ones with the highest local uni-read counts among the candidate positions of the multi-read within each replicate, i.e., comparing the local uni-read counts of all the available assignment positions for the multi-read within a replicate. This analysis indicated that, at all the trimmed read lengths, only a small fraction of the multi-reads (less than 12% for trimmed read length of 36bp and less than 4% for other read lengths and these percentages correspond to 1.45% and 0.76% of all the multi-reads in their respective settings), that were incorrectly assigned to the same location across the four replicates, were not assigned to the location with the maximum local uni-read signal. For the rest of the incorrectly assigned multi-reads, the final allocation position either had the highest local uni-read signal (56% to 61% of the incorrectly assigned multi-reads across all the trimmed read lengths) or had maximum local uni-read signal of zero.

**Author response image 1. respfig1:** Distribution of numbers of distinct assignment positions for incorrectly assigned multi-reads across 4 replicates of A549 at 40 kb resolution. mHi-C utilizes the trimmed uni-reads individually from four replicates to assign the same trimmed multi-reads from replicate 2 in the analysis of each trimmed replicate experiment. Each multi-read is either assigned to its true origin or can be incorrectly assigned to up to four different positions. Multi-reads that were incorrectly assigned to the same position are further stratified into three categories as: (i) assigned to the candidate position with the highest uni-reads signal (pink) (7.77% of all the multi-reads for trimming at 36bp); (ii) not assigned to the highest uni-reads enriched position (purple) (1.45% of all the multi-reads for trimming at 36bp); (iii) none of the candidate positions have any uni-reads (orange) (4.64% of all the multi-reads for trimming at 36bp).

The TAD analysis is still not very convincing (Figure 7B). The contact matrices look really similar between all conditions, but the called TADs for the Trimmed Uni-reads looks different for some reason. For example, the big TAD on the upper left doesn't seem to adhere to the likely TAD in the contact map. This may be a quirk of the TAD identification algorithm, but it doesn't make for a convincing visual case. The contact map intensity for many of the figures is getting clipped at local interactions (i.e. the region near the diagonal is essentially the same color of red), making it hard to see where the key differences are that drive the difference in the TAD calling. I would highly recommend either zooming in on smaller regions and/or changing the color scheme/scaling so that the TAD calls and the contact map can be more clearly evaluated by the reader.

Thanks for pointing this out. We updated Figure 7B and Figure 7—figure supplements 4-7 to better highlight the discrepancies in the contact matrices due to the exclusion of multi-reads and the resulting impact on boundary detection. In addition, we adjusted the color scale and zoomed in the region to highlight the major differences. The low sequencing depth in the trimmed Uni-setting does have a profound effect on the detection of TAD boundaries.

Reviewer #3:The authors have addressed most of my concerns. Two questions remain:1) Figure 4A: I am confused. First of all, the second track shouldn't be there – that's the whole chromosome. I think what the authors intended to plotted is the region in the tiny blue box. Second comment about this figure: if you look at the first track (virtual 4C), in the middle, there is a strong Uni&Multi peak, but when you look at the bottom two tracks (arcs), it shows it is uni-specific interaction. Can author clarify what happened?

We have removed the second track of the complete genome in Figure 4A and Figure 4—figure supplements 1 and 2. The following clarification addresses the second comment. The last track in Figure 4A contains all the significant interactions under Uni-setting and thus includes the ones that are common between the Uni- and Uni&Multi-settings. Originally, the second to the last track highlights all the novel significant interactions identified under Uni&Multi-setting, apart from those in the last track. Therefore, there is also a significant interaction from the region with a strong Uni&Multi-peak in the middle of the first track under Uni&Multi-setting. For clarity, we have changed the second to the last track into Uni&Multi-setting and therefore the ones not in the last track are novel promoter-enhancer interactions.

2) Regarding this response about its performance on normalized Hi-C matrix, the authors commented that "We updated all the visualizations to display both the normalized and unnormalized versions (Figure 2A, Figure 2—figure supplements 1-4). While the normalized Uni-setting matrices look less incomplete highlighting the impact of normalization, mHi-C generated matrices look visually more complete and display the impact of multi-reads. We provided several such examples as supplementary figures."I found the answer is not satisfactory – the authors need to provide quantitative evaluation of this claim.

We extended the original coverage analysis and directly quantified the difference between the normalized contact matrices under the Uni- and Uni&Multi-settings to highlight coverage improvement due to multi-reads.

The following paragraph, included in the main text, summarizes this quantification.

“Quantitatively, for the combined replicates of GM12878, 99.61% of the 5 kb bins with interaction potential are covered by at least 100 raw contacts under the Uni&Multi-setting, compared to 98.72% under Uni-setting, thereby allowing us to study 25.55Mb more of the genome. […] This also highlights that multi-reads alleviate the inflation of low raw contact count regions due to normalization. These major improvements in coverage provide direct evidence that mHi-C is rescuing multi-reads that originate from biologically valid fragments.”